# DPCore: Dynamic Prompt Coreset for Continual Test-Time Adaptation

**Yunbei Zhang** [1]  **Akshay Mehra** [2]  **Shuaicheng Niu** [3]  **Jihun Hamm** [1]

## Abstract

Continual Test-Time Adaptation (CTTA) seeks to adapt source pre-trained models to continually changing, unseen target domains. While existing CTTA methods assume structured domain changes with uniform durations, real-world environments often exhibit dynamic patterns where domains recur with varying frequencies and durations. Current approaches, which adapt the same parameters across different domains, struggle in such dynamic conditions—they face convergence issues with brief domain exposures, risk forgetting previously learned knowledge, or misapplying it to irrelevant domains. To remedy this, we propose **DPCore**, a method designed for robust performance across diverse domain change patterns while ensuring computational efficiency. DPCore integrates three key components: Visual Prompt Adaptation for efficient domain alignment, a Prompt Coreset for knowledge preservation, and a Dynamic Update mechanism that intelligently adjusts existing prompts for similar domains while creating new ones for substantially different domains. Extensive experiments on four benchmarks demonstrate that DPCore consistently outperforms various CTTA methods, achieving state-of-the-art performance in both structured and dynamic settings while reducing trainable parameters by 99% and computation time by 64% compared to previous approaches. Code is available at https://github.com/yunbeizhang/DPCore.

## 1. Introduction

Despite remarkable advances in deep learning, Deep Neural Networks (DNNs) often face significant performance

[1]Tulane University [2]Dolby Laboratories [3]Nanyang Technological University. Correspondence to: Yunbei Zhang <yzhang111@tulane.edu>, Jihun Hamm <jhamm3@tulane.edu>.

*Proceedings of the 42nd International Conference on Machine Learning*, Vancouver, Canada. PMLR 267, 2025. Copyright 2025 by the author(s).

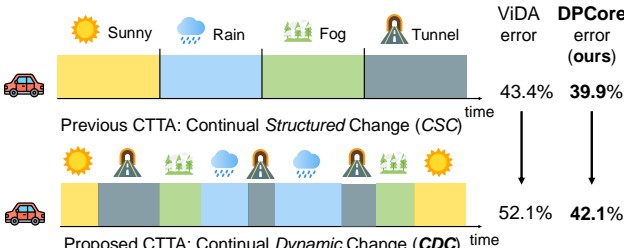

*Figure 1.* Illustrated through an autonomous driving scenario where a vehicle encounters varying weather and lighting conditions. The top panel shows the conventional CSC setting with structured, uniform-length domain transitions, while the bottom panel illustrates our proposed CDC setting where domains recur with varying frequencies and durations—better reflecting real-world challenges. When evaluated on ImageNet-to-ImageNet-C with ViT base model, previous SOTA ViDA's error rate increases significantly from 43.4% to 52.1% when moving from CSC to CDC, while DPCore maintains robust performance (39.9% to 42.1%).

degradation when encountering domain discrepancies between training and test environments (Recht et al., 2019; Hendrycks & Dietterich, 2019; Koh et al., 2021). This challenge, ubiquitous in real-world applications, has spurred the development of Test-Time Adaptation (TTA) (Wang et al., 2021; Iwasawa & Matsuo, 2021; Liang et al., 2020; Gao et al., 2023; Schneider et al., 2020; Mirza et al., 2022; 2023; Niu et al., 2024; Liang et al., 2023; Sun et al., 2020). TTA adapts pre-trained models to unseen target domains at test time without altering the original training process, making it particularly suitable for practical applications.

Real-world scenarios, however, present an even more challenging problem: continuously changing target domains. Consider an autonomous driving system that must rapidly adapt to varying weather conditions, lighting changes, and road conditions. Continual Test-Time Adaptation (CTTA) (Wang et al., 2022a; Niu et al., 2022) has emerged to address this challenge. The initial CTTA setting (Wang et al., 2022a), which we term Continual Structured Change (**CSC**), assumed domains change in a structured manner with uniform durations. However, real-world scenarios exhibit far more dynamic patterns where domains may recur multiple times with varying durations, from brief encounters to extended periods. We term this more realistic scenario Contin-

ual Dynamic Change (**CDC**). For instance, consider an autonomous vehicle driving through a mountainous highway, where it frequently transitions between sunny conditions, dark tunnels, dense fog in valleys, and sudden rain showers due to high humidity, as illustrated in the bottom of Fig. 1. Our experiments show that state-of-the-art (SOTA) CTTA methods like ViDA (Liu et al., 2023b) suffer substantial degradation in performance, with error rates increasing by 8.7% (from 43.4% to 52.1%) on ImageNet-to-ImageNet-C when moving from CSC to CDC settings.

When encountering CDC environments, existing CTTA methods, which typically adapt the same parameters across different domains (Wang et al., 2021; Liu et al., 2023b; Song et al., 2023), face three critical limitations. First, *Convergence Issue*: these methods inject numerous parameters that require substantial data to converge. While this is achievable in CSC where each domain persists for a long duration, it becomes problematic in CDC where certain domains appear only briefly. As shown in Fig. 4b, the increasing error rates during initial adaptation demonstrate that these methods struggle to achieve proper convergence with limited domain exposure. Second, *Catastrophic Forgetting*: since these methods continuously update the same large parameter space across all domains, knowledge learned from previous domains is progressively overwritten through successive adaptations, resulting in long-term performance degradation on previously encountered domains. Third, *Negative Transfer*: knowledge learned from one domain can adversely affect adaptation to substantially different domains, causing significant performance degradation (see in Sec. 4.3). While these issues exist in CSC, they are amplified in the CDC setting where domains change rapidly and unpredictably.

Furthermore, the practical deployment of CTTA models demands computational and memory efficiency, particularly for edge devices. Yet existing approaches overlook this requirement. Some methods rely on computationally intensive teacher-student architectures and extensive data augmentation (Wang et al., 2022a; Liu et al., 2023b), while others introduce large numbers of additional parameters that require pre-adaptation warm-up using source data (Liu et al., 2023b; Lee et al., 2024; Song et al., 2023; Gan et al., 2023).

To address these challenges, we introduce **D**ynamic **P**rompt **Core**set (**DPCore**), a novel CTTA method designed for robust performance across varying domain change patterns while maintaining computational efficiency. As shown in Fig. 2, DPCore combines three key components: (1) *Visual Prompt Adaptation* that aligns source and target domains with minimal parameters and no warm-up requirements, (2) a *Prompt Coreset* that preserves knowledge from previous domains while accommodating new ones, and (3) a *Dynamic Update* mechanism that intelligently adjusts prompts based on domain similarities. This design enables efficient learn-

ing from homogeneous domain groups while preventing negative knowledge transfer between dissimilar domains. Notably, DPCore requires only 1% of ViDA's parameters and merely 300 unlabeled source examples for computing statistics, eliminating the need for expensive warm-up procedures on the entire source dataset (see Table 4 and Fig. 5c).

Our extensive experiments across four CTTA benchmarks demonstrate DPCore's superiority. On ImageNet-to-ImageNet-C, DPCore achieves a +15.9% improvement over the source model, surpassing ViDA by 3.5% in the conventional CSC setting. Moreover, in the challenging CDC setting, DPCore maintains robust performance with an error rate of 42.1%, outperforming ViDA by 10.0%. This effectiveness extends to semantic segmentation, where DPCore improves mIoU scores over ViDA by 0.6% and 1.8% in CSC and CDC settings respectively. Our contributions are summarized as follows:

- We introduce a new CTTA setup—Continual Dynamic Change (CDC)—that better reflects real-world scenarios through frequent domain shifts and varying durations. We highlight that previous SOTA CTTA methods suffer significantly in CDC setup due to convergence issues, catastrophic forgetting, and negative transfer.

- We propose DPCore, a dynamic CTTA approach that effectively manages domain knowledge through a dynamic prompt coreset: preserving knowledge from seen domains, updating prompts for similar domains, and incorporating new knowledge for unseen domains.

- We show that DPCore is theoretically sound and achieves SOTA results for classification and segmentation tasks in both CSC and CDC settings while being computationally efficient.

## 2. Related Works

**Test-Time Adaptation (TTA)** (Wang et al., 2021; Iwasawa & Matsuo, 2021; Liang et al., 2020; Gao et al., 2023; Schneider et al., 2020; Mirza et al., 2022; 2023; Niu et al., 2024; Liang et al., 2023; Sun et al., 2020; Xiao & Snoek, 2024; Xiao et al., 2023; Mehra et al., 2024b) enhances pre-trained model performance using unlabeled data at test time, without access to the original training phase. TTA techniques fall into two main categories based on their use of source data. The first adjusts models through self-supervised losses like entropy minimization (Wang et al., 2021; Niu et al., 2023) and consistency maximization (Wang et al., 2022a; Liu et al., 2023b). The second involves preliminary steps using source data: either by extracting source characteristics such as statistics or features (Mirza et al., 2023; Niu et al., 2024; Zhang et al., 2024)) or by warming up injected parameters on source data before adaptation (Lee et al., 2024; Song et al., 2023; Liu et al., 2023b; Gao et al., 2023).

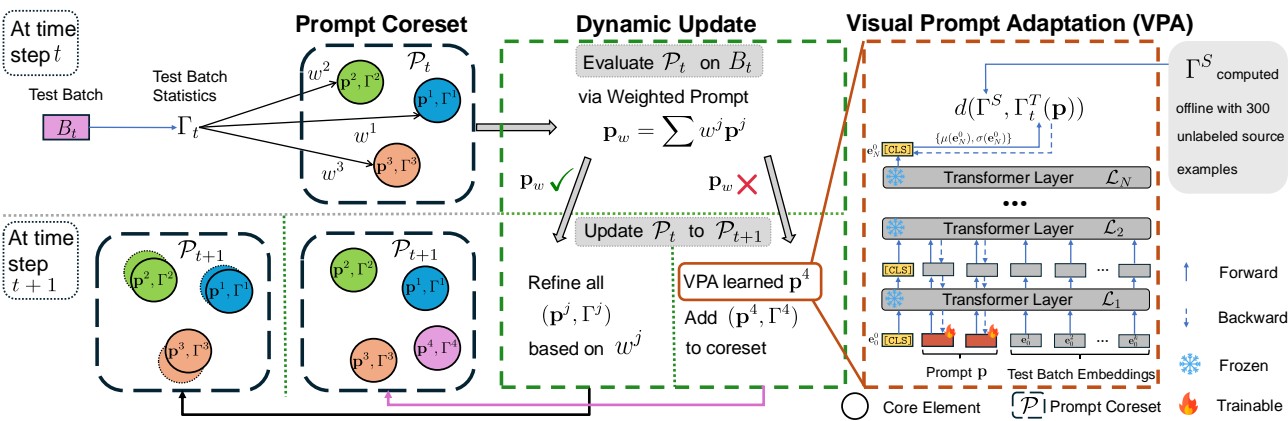

**Figure 2.** Overview of DPCore with three key components. At time step $t$, Prompt Coreset $\mathcal{P}_t$ (upper left) maintains core elements consisting of learned prompts $\boldsymbol{p}^j$ and statistics $\Gamma^j$ from previous domains. Dynamic Update (middle) evaluates new test batch $B_t$ by computing a weighted prompt $\boldsymbol{p}_w$ based on distances between batch statistics $\Gamma_t$ and core elements $\Gamma^j$. If $\boldsymbol{p}_w$ performs well, existing core elements $\{\boldsymbol{p}^j, \Gamma^j\}$ are refined using weights $w^j$; otherwise, Visual Prompt Adaptation (right) learns a new prompt by aligning the test batch with source statistics (computed offline from 300 examples) and adds it to the prompt coreset. The updated coreset $\mathcal{P}_{t+1}$ is then used for the next time step $t + 1$.

Crucially, both approaches are source-free adaptations since they operate without source data during adaptation. While our method falls into the second category, it significantly reduces source dependency compared to existing methods: DPCore requires only 300 unlabeled source examples to compute statistics, in contrast to DePT (Gao et al., 2023), ViDA (Liu et al., 2023b), VDP (Gan et al., 2023) and Be-CoTTA (Lee et al., 2024) which need the entire source dataset for warm-up. Moreover, as demonstrated in Fig. 5c, our approach maintains effectiveness even when source data is completely unavailable.

**Continual Test-Time Adaptation (CTTA)** (Wang et al., 2022a; Niu et al., 2022; Niloy et al., 2024; Liu et al., 2023a; Boudiaf et al., 2022; Yuan et al., 2023; Hoang et al., 2024; Press et al., 2023) tackles the challenge of continually changing target domains. This paradigm was first introduced by CoTTA (Wang et al., 2022a), which we refer to as Continual Structured Change (CSC), where domains change in a structured manner with clear boundaries and uniform lengths. While several subsequent works (Niu et al., 2023; Boudiaf et al., 2022; Gong et al., 2022; Yuan et al., 2023; Lee et al., 2024) explore TTA challenges such as mixed domains, label imbalance, temporal correlation, and gradual shifts, they still primarily focus on structured domain progressions or static domain scenarios. However, real-world environments often exhibit more dynamic patterns where domains recur with varying frequencies and durations—a critical aspect that existing setups often overlook. To address this gap, we introduce the Continual Dynamic Change (CDC) setting, where domains can appear multiple times with irregular durations, better reflecting realistic deployment scenarios.

The dynamic nature of CDC environments exposes funda-

mental limitations in current CTTA approaches (see Fig. 4b). Most existing methods employ a single-model strategy, updating the same set of parameters across all encountered domains (Wang et al., 2022a; Liu et al., 2023b; Hoang et al., 2024; Niu et al., 2022). While this approach works reasonably well in structured settings, it struggles significantly in dynamic environments due to three critical issues: convergence difficulties with brief domain exposures, catastrophic forgetting of previously learned knowledge, and negative transfer between dissimilar domains. Although some alternatives exist—such as PeTTA (Hoang et al., 2024), which uses memory components, or RDumb (Press et al., 2023), which employs periodic resets—these still fundamentally rely on single-model adaptation. In contrast, our DPCore addresses these limitations through a *multi-modeling* strategy via a dynamic prompt coreset. This approach learns and manages distinct prompts for different domain groups, enabling DPCore to preserve and reuse knowledge for similar domains while isolating learning for novel ones, thus providing robust adaptation across the varied domain changes characteristic of CDC settings.

## 3. Preliminaries and Problem Formulation

In this section, we introduce Vision Transformers (ViTs) and formally define the CTTA problem across both CSC and our proposed CDC settings.

### 3.1. Vision Transformers (ViTs)

We focus primarily on Vision Transformers (ViTs) due to their exceptional representation learning capabilities (Dosovitskiy et al., 2021; Liu et al., 2021). A ViT (Dosovitskiy

et al., 2021; Liu et al., 2021) $f$ with $N$ transformer layers $\{\mathcal{L}_i\}_{i=1}^N$ can be decomposed into a feature extractor $\phi : \mathcal{X} \to \mathcal{Z}$ with parameter $\theta_\phi$ and a classifier $h : \mathcal{Z} \to \mathcal{Y}$ with parameter $\theta_h$, such that $f = h \circ \phi$. Let $\boldsymbol{E}_i = \{\boldsymbol{e}_i^j\}_{j=0}^k$ denote the input sequence to the $(i+1)$-th layer $\mathcal{L}_{i+1}$, where $k$ is the number of image patches and $\boldsymbol{e}_i^0$ represents the classification token `[CLS]` from layer $\mathcal{L}_i$. The standard prediction process follows:

$$\boldsymbol{E}_i = \phi(\boldsymbol{E}_{i-1}), \ i = 1, ..., N \,, \tag{1}$$

$$\hat{y} = h(\boldsymbol{e}_N^0) \,. \tag{2}$$

### 3.2. CTTA Problem Formulation

Given a model $f_\theta$ with parameters $\theta$ pre-trained on source domain $\mathcal{D}^S = (X^S, Y^S)$, the goal of Continual Test-Time Adaptation (CTTA) is to adapt this model to a sequence of unlabeled target domains $\{\mathcal{D}^{T_1}, \mathcal{D}^{T_2}, ..., \mathcal{D}^{T_M}\}$, where $M$ denotes the number of potential target domains and can be unknown or infinite in real-world scenarios. The model $f_\theta$ operates in an online setting where it processes a sequence of test data batches $\{B_t\}_{t=1}^\infty$, encountering one batch $B_t$ at each time step $t$. Following standard assumptions in the literature (Wang et al., 2022a; Yang et al., 2024), we consider all samples within a batch $B_t$ belong to the same target domain, though the domain identity remains unknown. At each time step $t$, CTTA aims to adapt the model parameters from $\theta_t$ to $\theta_{t+1}$ by learning from the current batch $B_t$, to improve prediction performance on future batches.

### 3.3. Continual Dynamic Change: A new CTTA setup

The CTTA setting in previous works (Wang et al., 2022a; Liu et al., 2023b; 2024) typically assumes that target domains change in a structured manner: each domain has uniform length and changes occur at regular intervals (Fig. 1 Top). However, real-world scenarios present more dynamic patterns where the same domain may appear multiple times with varying durations, (Fig. 1 Bottom). To systematically study this setting, we simulate dynamic environments using a Dirichlet distribution with parameter $\delta$. As demonstrated in Fig. 4a, smaller values of $\delta$ result in domain transitions more closely resembling the conventional CSC setting. As $\delta$ increases, domain changes become more frequent and unpredictable. For our main experiments, we set $\delta = 1$ to keep a balance between structured and completely random domain changes. Various $\delta$ and other distributions are discussed in Appendix D.

## 4. Method

In this section, we present Dynamic Prompt Coreset (**DPCore**), illustrated in Fig. 2, consisting of three components: *Visual Prompt Adaptation*, *Prompt Coreset*, and *Dynamic Update*. These components work together to effec-

tively manage domain knowledge by preserving knowledge from previously visited domains, updating learned prompts when encountering similar domains, and incorporating new knowledge when facing new domains.

### 4.1. Visual Prompt Adaptation (VPA)

We leverage visual prompts (Jia et al., 2022; Ge et al., 2023) for efficient adaptation at test time by introducing $L$ learnable tokens $\boldsymbol{p} := \{[\texttt{Prompt}]_i\}_{i=1}^L$, where $[\texttt{Prompt}] \in \mathbb{R}^{768}$ for ViT-Base models. These prompts augment the input sequence as $\boldsymbol{E}_i' = \{\boldsymbol{e}_i^0, \boldsymbol{p}, \boldsymbol{e}_i^1, ..., \boldsymbol{e}_i^k\}$, modifying Eq. (1) to:

$$\boldsymbol{E}_i' = \phi(\boldsymbol{E}_{i-1}'), \ i = 1, ..., N \,. \tag{3}$$

As illustrated on the right of Fig. 2, we first extract source features $\mathcal{Z}^S = \phi(\mathcal{X}^S; \theta_\phi)$, i.e., $\boldsymbol{e}_N^0$, in a one-time offline computation before adaptation. For the current batch $B_t^T$ at time step $t$ from unknown target domain $T$, we initialize prompt tokens from a Gaussian distribution (Jia et al., 2022) without any warm-up. Denote the extracted features of the current batch with prompt $\boldsymbol{p}$ by $\mathcal{Z}_t^T(\boldsymbol{p}) := \phi(B_t^T; \theta_\phi, \boldsymbol{p})$. We align the statistics (mean and standard deviation) of $B_t^T$, $\Gamma_t^T(\boldsymbol{p}) := \{\boldsymbol{\mu}_t^T(\boldsymbol{p}), \boldsymbol{\sigma}_t^T(\boldsymbol{p})\} := \{\boldsymbol{\mu}(\mathcal{Z}_t^T(\boldsymbol{p})), \boldsymbol{\sigma}(\mathcal{Z}_t^T(\boldsymbol{p}))\}$ with the source $\Gamma^S(\boldsymbol{p}) := \{\boldsymbol{\mu}^S, \boldsymbol{\sigma}^S\} := \{\boldsymbol{\mu}(\mathcal{Z}^S), \boldsymbol{\sigma}(\mathcal{Z}^S)\}$ through the following distance

$$d(\Gamma^S, \Gamma_t^T(\boldsymbol{p})) = \|\boldsymbol{\mu}^S - \boldsymbol{\mu}_t^T(\boldsymbol{p})\|_2 + \|\boldsymbol{\sigma}^S - \boldsymbol{\sigma}_t^T(\boldsymbol{p})\|_2 \,. \tag{4}$$

The prompt is then learned from scratch by minimizing this distance

$$\boldsymbol{p}^* = \arg\min_{\boldsymbol{p}} \ d(\Gamma^S, \Gamma_t(\boldsymbol{p})) \,, \tag{5}$$

where we omit the superscript $T$ for simplicity, and the subscript $t$ denotes the test batch $B_t$. Note that while this distance requires source data, labels are unnecessary since we only perform marginal distribution alignment. Our empirical results in Fig. 5c show that merely 300 unlabeled source examples are enough to achieve stable performance.

### 4.2. Prompt Coreset

To address catastrophic forgetting in continually changing environments (Niu et al., 2022; Kirkpatrick et al., 2017; Li & Hoiem, 2017), we introduce a Prompt Coreset mechanism inspired by Online K-Means (Duda & Hart, 1973). The coreset is initialized as an empty set $\mathcal{P}_0$ as adaptation begins and undergoes adaptive updates at each time step $t$: $\mathcal{P}_t \to \mathcal{P}_{t+1}$. Each core element in the coreset consists of a learned prompt and its corresponding feature statistics, illustrated on the left of Fig.2. For example, when the first batch $B_1$ arrives, we begin by extracting features using the pre-trained model $f_\theta$ without any prompt, denoted as $\mathcal{Z}_1$, and compute its statistics $\Gamma_1 = \{\boldsymbol{\mu}_1, \boldsymbol{\sigma}_1\} = \{\boldsymbol{\mu}(\mathcal{Z}_1), \boldsymbol{\sigma}(\mathcal{Z}_1)\}$. We use $\mathcal{Z}_1(\boldsymbol{p})$ and $\mathcal{Z}_1$ to distinguish features extracted with

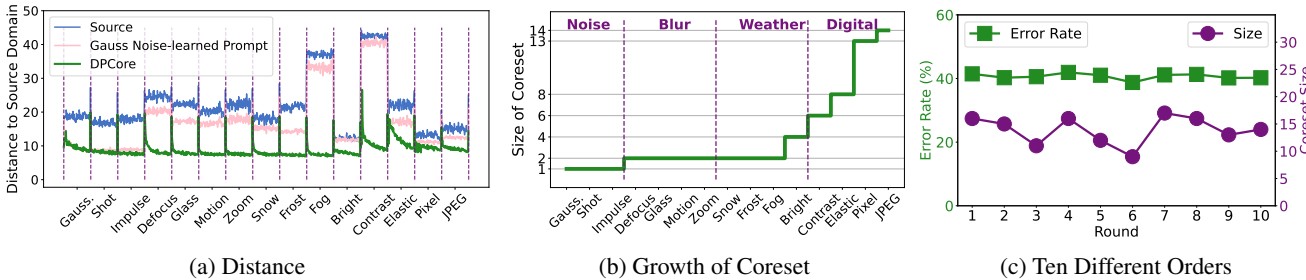

(a) Distance  (b) Growth of Coreset  (c) Ten Different Orders

*Figure 3.* Analysis of DPCore on ImageNet-to-ImageNet-C in the CSC setting. (a) Distance between test batches and source domain in Eq. (4) using source model (baseline), DPCore, and a prompt trained only on Gaussian Noise. DPCore consistently reduces domain gaps across all corruptions. (b) Evolution of coreset size across corruption types, showing strategic grouping within the four main corruption categories (Noise, Blur, Weather, Digital). (c) Performance stability across ten different domain orders.

and without prompts respectively. Subsequently, we learn a prompt $\boldsymbol{p}_1$ from scratch for $B_1$ through Eq. (5). The learned prompt and its corresponding statistics form a pair $(\boldsymbol{p}_1, \Gamma_1)$ that is stored in the coreset as its first element. Unlike existing approaches such as (Wang et al., 2022b) that require maintaining a buffer of test data, our method stores only feature statistics, significantly reducing memory requirements while enhancing data privacy protection.

### 4.3. Dynamic Update to the Prompt Coreset

**Motivations.** VPA has demonstrated effectiveness in single-domain TTA (Niu et al., 2024; Zhang et al., 2024). Our empirical studies in Appendix Table 6 show that prompts learned from one domain can significantly benefit adaptation to similar domains (e.g., a Gaussian Noise-learned prompt achieves 40.4% error rate on Shot Noise, comparable to a Shot Noise-learned prompt's 39.9%). However, this transferability is not universal as the same prompts can be ineffective or even harmful for substantially different domains (e.g., increasing error rate from 91.4% to 95.7% on Contrast). These findings suggest that existing prompts can be effective for similar domains while new ones should be learned for significantly different domains. To achieve this, we employ a dynamic update strategy which first evaluates the current coreset on the test batch and updates the prompt coreset based on the evaluation result.

**Evaluating $\mathcal{P}_t$ on $B_t$ via Weighted Prompt.** We employ a distance-based method to evaluate each new batch after the first batch against existing core elements. Specifically, at time step $t > 1$, given $K > 0$ elements in the coreset $\{(\boldsymbol{p}^j, \Gamma^j)\}_{j=1}^K$, we first extract features $\mathcal{Z}_t$ from $B_t$ without any prompt, i.e., $\mathcal{Z}_t = \phi(B_t; \theta_\phi)$. and compute its statistics $\Gamma_t$. Here, we abuse the superscript to note the element in the coreset. Rather than evaluating each core element individually, which would require processing the same batch $K$ times, we employ a distance-based weighted prompt approach and process $B_t$ with the weighted prompt only

once. DPCore computes distances $d(\Gamma_t, \Gamma^j)$ between the current batch statistics and all existing core elements, then generates a weighted composition of all $K$ prompts $\boldsymbol{p}_w$ as

$$\boldsymbol{p}_w := \sum_{j=1}^K w^j \boldsymbol{p}^j,$$

$$\text{where } w^j = \frac{\exp(-d(\Gamma_t, \Gamma^j)/\tau)}{\sum_{l=1}^K \exp(-d(\Gamma_t, \Gamma^l)/\tau)} \quad (6)$$

and $\tau$ is a temperature parameter. The weighted prompt approach offers more flexibility than using the nearest neighbor. When a new domain shares characteristics with multiple visited domains, it can be effectively decomposed into a weighted combination of existing prompts, providing more accurate adaptation than forcing alignment with a single most similar domain. Notably, all statistics used here are based on features extracted without prompts, as the source pre-trained model can effectively evaluate sample similarity (Yang et al., 2024) while requiring less computation.

**Updating $\mathcal{P}_t$ to $\mathcal{P}_{t+1}$.** We evaluate the effectiveness of $\boldsymbol{p}_w$ by comparing the distances without and with weighted prompts $d(\Gamma^S, \Gamma_t)$ and $d(\Gamma^S, \Gamma_t(\boldsymbol{p}_w))$. If the distance reduction is insufficient, indicating $B_t$ likely comes from a new domain, we learn a new prompt from scratch using Eq. (5) and add the pair $(\Gamma_t, \boldsymbol{p}_t)$ to the coreset as a new element, following the same procedure used for the first batch $B_1$. Conversely, if the distance decreases significantly, indicating similarity to previously seen domains, we refine $\boldsymbol{p}_w$ with a single update step to obtain the final prompt $\boldsymbol{p}_t$ for $B_t$. The core elements are then adaptively updated based on their weights $w^j$ and an updating weight $\alpha$ as

$$\boldsymbol{p}^j \leftarrow \boldsymbol{p}^j + \alpha w^j(\boldsymbol{p}_t - \boldsymbol{p}^j), \ \Gamma^j \leftarrow \Gamma^j + \alpha w^j(\Gamma_t - \Gamma^j). \quad (7)$$

To ensure valid statistics, we use $\max\{\mathbf{0}, (\boldsymbol{\sigma}_t - \boldsymbol{\sigma}^i)\}$ in the update to prevent negative standard deviations. We compute a ratio of the two distances $\frac{d(\Gamma^S, \Gamma_t(\boldsymbol{p}_w))}{d(\Gamma^S, \Gamma_t)}$ and compare it with a predefined threshold ratio $\rho$ to evaluate whether a

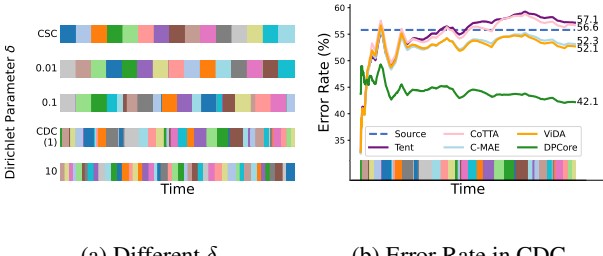

(a) Different $\delta$      (b) Error Rate in CDC

*Figure 4.* Analysis of CDC setting on ImageNet-to-ImageNet-C. (a) Domain change patterns with varying Dirichlet distribution parameter $\delta$ (colors represent different domains): smaller $\delta$ produces CSC-like structured changes while larger $\delta$ leads to more frequent and unpredictable transitions. (b) Performance comparison in CDC setting ($\delta = 1$), where previous methods show significant degradation while DPCore maintains robust performance.

batch represents a potential new domain. The ratio is used for applying a domain-dependent distance threshold. This approach allows the coreset to grow dynamically based on the complexity of the domain space, demonstrated in Fig. 3b. Given that the total number of domains is typically unknown in real-world scenarios, we do not impose strict constraints on the coreset size, allowing it to adapt naturally to the complexity of the domain space.

The strength of our algorithm lies in two key aspects. First, it learns prompts from homogeneous domain groups rather than individual batches, enabling more comprehensive adaptation. This ability is particularly valuable when domains change rapidly with short durations—where previous CTTA methods might struggle with convergence, our method can effectively learn and utilize prompts even across discontinuous domain appearances. Second, it mitigates error accumulation by avoiding updates to existing prompts when encountering significantly different domains. The complete algorithm is presented in Appendix Alg. 1.

### 4.4. Theoretical Analysis

We further provide theoretical insight into why DPCore maintains consistent performance across domain changes. Inspired by online K-Means (Duda & Hart, 1973), we analyze a simplified version of DPCore where batches naturally cluster into distinct groups based on their distances. Under the assumption that these clusters are well-separated, we prove three key propositions: 1) (Prop B.1) DPCore correctly assigns batches to their respective clusters regardless of the sequence length, 2) (Prop B.2) these assignments remain correct regardless of the order in which batches arrive, and 3) (Prop B.3) the learned prompts are independent of batch order. Together, these properties demonstrate that DPCore can effectively manage domain knowledge even when domains appear in arbitrary orders with varying fre-

quencies—a crucial advantage in dynamic environments. Detailed analysis is provided in Appendix B.4.

## 5. Experiments

In this section, we evaluate the effectiveness of DPCore on classification and segmentation tasks across CSC and CDC settings. Our evaluation focuses on three aspects: 1) its ability to handle (dynamically) changing domains, 2) the growth of coreset, and 3) its hyperparameter sensitivity.

### 5.1. Experimental Setup

**Datasets.** We evaluate our proposed method on three classification datasets: ImageNet-to-ImageNet-C, CIFAR100-to-CIFAR100C, and CIFAR10-to-CIFAR10C (Hendrycks & Dietterich, 2019). Each dataset comprises corrupted images in 15 types of corruption, categorized into four main groups: Noise, Blur, Weather, and Digital. Our experiments use the highest level of corruption (i.e., severity 5). Moreover, we validate DPCore on a CTTA segmentation task: Cityscapes-ACDC (Sakaridis et al., 2021; Cordts et al., 2016).

**Methods Compared.** We compare our method against strong CTTA baselines including Tent (Wang et al., 2021), SAR (Niu et al., 2023), CoTTA (Wang et al., 2022a), VDP (Gan et al., 2023), DePT (Gao et al., 2023), RoTTA (Yuan et al., 2023), EcoTTA (Song et al., 2023), ViDA (Liu et al., 2023b), BDG (Yang et al., 2024), and C-MAE (Liu et al., 2024). Following their protocols, we conduct several warm-up iterations on labeled source data (e.g., ImageNet) for VDP, DePT, ViDA, and EcoTTA to ensure optimal performance. Implementation details are in Appendix A.

**CSC and CDC Settings.** We evaluate DPCore across the CSC and CDC settings. The CSC setting follows the implementation by (Wang et al., 2022a). In the experiment under CDC setting, we employ Dirichlet Distribution with $\delta = 1$ to generate the data stream from all domains dynamically. Other distributions are discussed in Appendix D.

**Implementation Details.** To ensure consistency and comparability in our experiments, we follow the implementation details specified in CoTTA (Wang et al., 2022a) and ViDA (Liu et al., 2023b). For classification tasks, we use the ViT-Base model (Wightman, 2019), while for segmentation tasks, we employ the pre-trained Segformer-B5 model (Xie et al., 2021). Following the protocol established in (Zhang et al., 2022), we determine hyperparameters using the four disjoint validation corruptions provided with ImageNet-C and CIFAR10-C (Hendrycks & Dietterich, 2019). We utilize the AdamW optimizer (Loshchilov & Hutter, 2017; Kingma & Ba, 2014) with a learning rate of 0.01, and select the number of prompt tokens at $L = 8$ and the threshold at $\rho = 0.8$. The source statistics are computed using 300 examples from the corresponding source domain. For other

*Table 1.* Classification error rate (%) for ImageNet-to-ImageNet-C in previous CSC setting, evaluated on ViT-Base with corruption severity level 5. **Bold** and underline indicate best and second-best performance respectively.

| Algorithm | Gauss. | Shot | Impulse | Defocus | Glass | Motion | Zoom | Snow | Frost | Fog | Bright | Contrast | Elastic | Pixel | JPEG | Mean↓ | Gain↑ |
|---|---|---|---|---|---|---|---|---|---|---|---|---|---|---|---|---|---|
| Source | 53.0 | 51.8 | 52.1 | 68.5 | 78.8 | 58.5 | 63.3 | 49.9 | 54.2 | 57.7 | 26.4 | 91.4 | 57.5 | 38.0 | 36.2 | 55.8 | 0.0 |
| Pseudo (Lee, 2013) | 45.2 | 40.4 | 41.6 | 51.3 | 53.9 | 45.6 | 47.7 | 40.4 | 45.7 | 93.8 | 98.5 | 99.9 | 99.9 | 98.9 | 99.6 | 61.2 | -5.4 |
| Tent (Wang et al., 2022a) | 52.2 | 48.9 | 49.2 | 65.8 | 73.0 | 54.5 | 58.4 | 44.0 | 47.7 | 50.3 | 23.9 | 72.8 | 55.7 | 34.4 | 33.9 | 51.0 | +4.8 |
| CoTTA (Wang et al., 2022a) | 52.9 | 51.6 | 51.4 | 68.3 | 78.1 | 57.1 | 62.0 | 48.2 | 52.7 | 55.3 | 25.9 | 90.0 | 56.4 | 36.4 | 35.2 | 54.8 | +1.0 |
| VDP (Gan et al., 2023) | 52.7 | 51.6 | 50.1 | 58.1 | 70.2 | 56.1 | 58.1 | 42.1 | 46.1 | 45.8 | 23.6 | 70.4 | 54.9 | 34.5 | 36.1 | 50.0 | +5.8 |
| SAR (Niu et al., 2023) | 45.8 | 45.9 | 47.7 | 52.3 | 63.7 | 46.2 | 50.9 | 40.3 | 42.4 | 41.8 | 24.4 | 53.4 | 53.6 | 38.4 | 36.6 | 45.6 | +10.2 |
| RoTTA (Yuan et al., 2023) | 51.5 | 50.3 | 51.7 | 60.4 | 58.7 | 52.6 | 54.8 | 47.2 | 43.5 | 42.8 | 25.9 | 49.1 | 48.8 | 46.3 | 39.7 | 48.2 | +7.6 |
| RDumb (Press et al., 2023) | 46.4 | 42.1 | 43.7 | 56.1 | 53.1 | 47.2 | 49.7 | 43.0 | 41.1 | 46.8 | 27.7 | 52.3 | 49.9 | 35.3 | 34.6 | 44.6 | +11.2 |
| EcoTTA (Song et al., 2023) | 48.1 | 45.6 | 46.3 | 56.5 | 67.1 | 50.4 | 57.1 | 41.3 | 44.5 | 43.8 | 24.1 | 71.6 | 54.8 | 34.1 | 34.8 | 48.0 | +7.8 |
| ViDA (Liu et al., 2023b) | 47.7 | 42.5 | 42.9 | 52.2 | 56.9 | 45.5 | 48.9 | 38.9 | 42.7 | 40.7 | 24.3 | 52.8 | 49.1 | 33.5 | 33.1 | 43.4 | +12.4 |
| BGD (Yang et al., 2024) | 47.5 | 42.1 | 41.6 | 55.5 | 55.4 | 44.5 | 47.9 | 38.8 | 37.8 | 39.6 | 23.6 | 57.0 | 44.4 | 33.5 | 32.3 | 42.8 | +13.0 |
| C-MAE (Liu et al., 2024) | 46.3 | 41.9 | 42.5 | 51.4 | 54.9 | 43.3 | 40.7 | 34.2 | 35.8 | 64.3 | 23.4 | 60.3 | 37.5 | 29.2 | 31.4 | 42.5 | +13.3 |
| **DPCore (Proposed)** | **42.2** | **38.7** | **39.3** | **47.2** | **51.4** | 47.7 | 46.9 | 39.3 | 36.9 | **37.4** | 22.0 | 44.4 | 45.1 | **30.9** | **29.6** | **39.9** | **+15.9** |

datasets that lack validation sets, we apply these same hyperparameters without additional tuning to align with practical testing conditions, where hyperparameters are selected before accessing any target data. More details of DPCore's implementation are available in Appendix B.3. We ensure a fair comparison by maintaining the same hyperparameters across both the CSC and CDC settings for all methods.

## 5.2. Main Results

We present experimental results on ImageNet-to-ImageNet-C and Cityscapes-to-ACDC across both CSC and CDC settings. **Results for CIFAR10/100-to-CIFAR10/100 are provided in Appendix C.3** due to space limit.

**ImageNet-to-ImageNet-C in CSC.** We first evaluated DPCore in the conventional CSC setting, where 15 types of corruption occur sequentially, from Gaussian Noise to JPEG compression during test time. Results, as shown in Table 1, indicate that DPCore achieves a SOTA improvement of +15.9% over the source model, surpassing the second-best method, C-MAE, by 2.6%. The coreset's evolution during this process, illustrated in Fig. 3b, reveals that DPCore does not treat each corruption type independently but rather groups similar corruptions to optimize adaptation. For instance, examining the Noise group (Gaussian, Shot, Impulse) in Fig. 3a, we observe that these domains share similar distances to the source domain. Recognizing this similarity, DPCore efficiently handles the entire group with a single core element. In addition, DPCore shows sophistication in handling domains that exhibit significant differences. A notable example occurs in the Weather group: when transitioning from Fog to Brightness, the distance metrics change dramatically despite both corruptions belonging to the same category. In response, DPCore adapts by learning two separate core elements for Brightness, leading to superior performance compared to other baselines on the Brightness domain. Overall, DPCore maintains efficiency by limiting the total number of core elements to 14 and demonstrates remarkable flexibility in distributing these elements across corruption types. This distribution is

strategically varied based on both the complexity of individual domains and their similarities to other corruptions, reflecting a sophisticated approach to domain management.

To further validate DPCore's robustness, we conducted additional experiments using 10 random domain orders, following the evaluation protocol of CoTTA (Wang et al., 2022a). DPCore demonstrates consistent performance, achieving an average error rate of 40.2% with an average coreset size of 13.9 elements in Fig. 3c. Details are available in Sec. C.2.

**ImageNet-to-ImageNet-C in CDC.** We benchmark CTTA methods in the more challenging CDC setting. Results in Fig. 4b reveal significant challenges for existing CTTA methods in this scenario. Although these methods perform well in CSC, their gains diminish substantially in the more realistic CDC environment. Strikingly, previous SOTA methods suffer severe degradation: ViDA's error rate increases dramatically from 43.4% to 52.1%, while Tent not only loses its advantage but performs worse than the source model, with error rates increasing from 51.0% to 57.1%, compared to the source model's 55.8%. These substantial performance drops highlight the limitations of existing methods when handling varying domain lengths and rapid domain shifts.

In contrast, DPCore shows remarkable resilience, experiencing only a modest increase in error rate from 39.9% to 42.1%, outperforming the second-best method by a significant margin of 10.0%. To handle the dynamic domain changes, DPCore adapts its coreset size from 14 to 25 elements between CSC and CDC settings. This allows DPCore to maintain performance stability across diverse scenarios.

**Cityscapes-to-ACDC in CSC and CDC.** Beyond classification, we evaluate DPCore on real-world semantic segmentation using the Cityscapes-to-ACDC dataset, where performance is measured by mean Intersection over Union (mIoU, %). As shown in Table 2, in the CSC setting following (Liu et al., 2023b; 2024) where domain sequences repeat three times, DPCore shows consistent improvement in mIoU across cycles (62.1 → 62.6 → 62.8), outperforming previous SOTA ViDA by 0.6%. In the more challenging CDC

*Table 2.* Average mIoU score (%) for Cityscapes-to-ACDC across both CSC and CDC settings. The same target domains are repeated three rounds. Full results in Appendix C.4.

| Setting | Algorithm | Round 1 | Round 2 | Round 3 | Mean↑ | Gain↑ |
|---|---|---|---|---|---|---|
| | Source | 56.7 | 56.7 | 56.7 | 56.7 | 0.0 |
| CSC | Tent | 56.7 | 55.9 | 55.0 | 55.7 | -1.0 |
| | CoTTA | 58.6 | 58.6 | 58.6 | 58.6 | +1.9 |
| | EcoTTA | 56.0 | 55.9 | 55.8 | 55.8 | -0.9 |
| | ViDA | 61.1 | 62.2 | 62.3 | 61.9 | +5.2 |
| | C-MAE | 61.8 | 61.6 | 62.0 | 61.8 | +5.1 |
| | **DPCore** | **62.1** | **62.6** | **62.8** | **62.5** | **+5.8** |
| CDC | Tent | 56.6 | 54.8 | 53.4 | 54.9 | -1.8 |
| | CoTTA | 58.0 | 57.3 | 56.5 | 57.3 | +0.5 |
| | EcoTTA | 55.6 | 55.3 | 55.1 | 55.3 | -1.4 |
| | ViDA | 60.1 | 59.1 | 58.6 | 59.2 | +2.5 |
| | C-MAE | 59.6 | 58.8 | 60.0 | 58.7 | +2.0 |
| | **DPCore** | **61.4** | **60.8** | **60.7** | **61.0** | **+4.3** |

*Table 3.* Effect of DPCore's three components: VPA (Visual Prompt Adaptation), PC (Prompt Coreset), and DU (Dynamic Update). Time shows relative computation time (Tent=1.0), Err Mean shows classification error rate (%).

| | VPA | PC | DU | Time | Err Mean↓ | Gain↑ |
|---|---|---|---|---|---|---|
| Source | - | - | - | - | 55.8 | - |
| Tent | - | - | - | 1.0 | 51.0 | +4.8 |
| Exp-1 | ✓ | - | ✓ | 1.0 | 50.8 | +5.0 |
| Exp-2 | ✓ | ✓ | - | 11.4 | 48.3 | +7.5 |
| Exp-3 | - | ✓ | ✓ | 1.6 | 45.1 | +10.7 |
| DPCore | ✓ | ✓ | ✓ | 1.8 | **39.9** | **+15.9** |

setting, while Tent and EcoTTA perform worse than the source model and ViDA's improvement drops from +5.2% to +2.5%, DPCore maintains robust performance (from +5.8% to +4.3%) while adapting its coreset size from 5 elements in CSC to 7 in CDC to handle increased domain complexity. Fine-grained results are available in Appendix C.4.

## 5.3. Ablation Studies

We provide comprehensive analyses and ablation studies for DPCore on ImageNet-to-ImageNet-C in the CSC setting. Additional results are available in Appendix E.

**Effect of Each Component.** Table 3 evaluates the contributions of DPCore's key components: Visual Prompt Adaptation (VPA), Prompt Coreset (PC), and Dynamic Update (DU). In the first experiment (Exp-1), employing only VPA and DU without PC, we see a 4.8% reduction in error rate compared to the source pre-trained model, with computation time similar to Tent as it involves learning a single prompt from scratch for all domains. Exp-2 introduces PC but omits DU, requiring new prompts for each batch, enhancing performance to +7.5% but increasing computation time tenfold compared to Tent. In Exp-3, we replace VPA with NormLayer parameters while retaining PC and DU, outperforming Tent by +5.9%, illustrating the method's adaptability beyond ViT architectures to CNNs by adapting NormLayer parameters. The full DPCore setup, integrating VPA, PC, and DU, achieves a SOTA improvement of +15.9% while maintaining computational efficiency.

**Effect of prompt length $L$.** We assess the impact of prompt size by varying the number of prompt tokens $L$ within the range of $\{1, 2, ..., 10\}$. As depicted in Fig. 5a, DPCore's performance exhibits strong stability across different values of $L$, demonstrating low sensitivity to this parameter. We fix $L = 8$ for all main experiments. **DPCore's sensitivity to threshold $\rho$.** The threshold $\rho$ in Alg. 1 controls DPCore's sensitivity to domain changes. Lower $\rho$ values result in more batches being classified as samples from unseen domains. For example, setting $\rho$ near zero causes each batch

to learn its own prompt. In contrast, high $\rho$ values lead to all batches after the first one being considered as from visited domains, potentially reducing the coreset to a single element. However, even a high ratio $\rho$ such as 1 is likely to learn a new core element if the weighted prompt $\boldsymbol{p}_w$ increases the domain gap significantly for different unseen domains, i.e., $d(\Gamma^S, \Gamma_t(\boldsymbol{p}_w)) > d(\Gamma^S, \Gamma_t)$. We examine $\rho$ within a range of 0.1 to 1.0. Experimental results, as shown in Fig. 5b, demonstrate stable performance for $\rho$ values from 0.6 to 0.9. To ensure consistency and avoid dataset-specific tuning, we fix $\rho = 0.8$ across all our main experiments before any exposure to target data.

**Effect of test batch size.** To comprehensively assess the impact of test batch size, we evaluate various CTTA methods across batch sizes ranging from 1 to 256. The results illustrated in Fig. 5d reveal a consistent pattern across methods: regardless of their objective functions (entropy, consistency, or distribution alignment), they maintain stable performance with sufficiently large batch sizes but exhibit increased error rates as batch sizes decrease. In the extreme case of single-sample adaptation (batch size one), all methods demonstrate significantly degraded performance, falling below the source model's baseline. This observation aligns with findings reported in recent literature (Yuan et al., 2023; Song et al., 2023). Based on these empirical findings and consistent with prior work, we standardize the batch size to 64 for fair comparison across methods. Moreover, DPCore maintains exceptional stability for batch sizes above 16 and shows minimal degradation for smaller sizes (8 and 4) while outperforming other methods. For extremely small batch sizes (e.g. 1), we introduce DPCore-B, which uses a buffer zone to accumulate samples before updates. With a buffer size of 64, DPCore-B achieves a 41.2% error rate even with single-sample batches, demonstrating its practical utility (details in Appendix E.1).

**Effect of number of source examples.** We evaluate DPCore's sensitivity to source data volume by varying the number of source examples used for computing statistics from 0 to 10k. As shown in Fig. 5c, DPCore maintains effective adaptation even with 50 examples. For all main experiments, we randomly select 300 unlabeled source examples, notably different from methods like VDP, DePT, EcoTTA, and ViDA

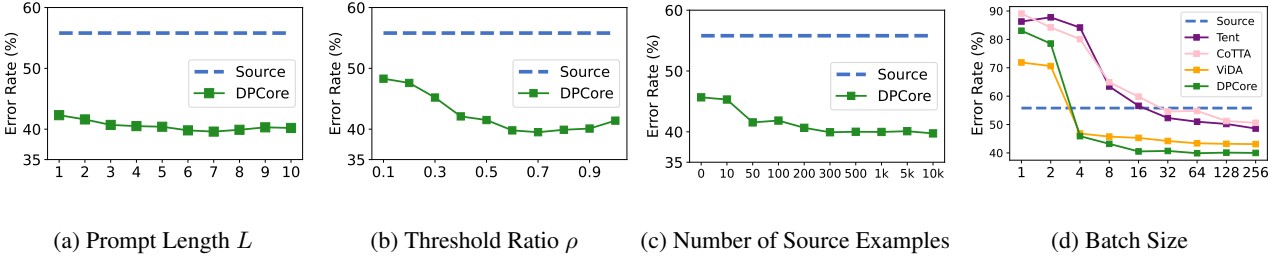

|                |                |                |                |
|:--------------:|:--------------:|:--------------:|:--------------:|
| (a) Prompt Length $L$ | (b) Threshold Ratio $\rho$ | (c) Number of Source Examples | (d) Batch Size |

*Figure 5.* Ablation studies on ImageNet-to-ImageNet-C: impact of (a) prompt token length, (b) threshold ratio for core element evaluation, (c) number of source examples used for statistics computation, and (d) batch size on classification error rate.

*Table 4.* Computational analysis on ImageNet-to-ImageNet-C. #Param. shows learnable parameters (Extra Param. indicates injected parameters beyond model), #FP/#BP shows propagation counts, Time shows relative computation (Tent=1.0), and Err Mean shows classification error rate (%).

| Algo. | #Param. | Extra Param. | #FP | #BP | Time | Err. Mean↓ |
|:-----:|:-------:|:------------:|:---:|:---:|:----:|:----------:|
| Tent | 0.03M | | 1 | 1 | 1.0 | 51.0 |
| CoTTA | 86.57M | | 11.7 | 1 | 3.6 | 54.8 |
| VDP | 1800 | ✓ | 2 | 1 | 1.5 | 50.0 |
| EcoTTA | 3.46M | ✓ | 1 | 1 | 1.9 | 48.0 |
| ViDA | 93.70M | ✓ | 11 | 1 | 2.8 | 43.4 |
| **Ours** | **0.08M** | ✓ | **3.1** | **1.1** | **1.8** | **39.9** |

*Table 5.* Memory management strategies for fixed-size coreset (K=15) on ImageNet-to-ImageNet-C across 10 different orders.

| Algo. | Source | DPCore | | |
|:-----:|:------:|:------------:|:--------------:|:-------------:|
|       |        | (flexible K) | (K=15, discard) | (K=15, merge) |
| **Err. Mean ↓** | 55.8 | 40.2 | 42.3 | 41.1 |

that require the entire labeled source dataset for parameter initialization. We further explore an extreme scenario where source data is completely inaccessible (shown at source data = 0 in Fig. 5c). For ImageNet-C experiments, using 300 unlabeled STL10 (Coates et al., 2011) images as reference still achieves an error rate of 45.6%, improving over the source model by 10.2%. Details are provided in Appendix E.2.

**DPCore's computation and memory efficiency.** We analyze computational complexity across methods in Table 4, comparing learnable parameters, forward/backward propagation counts, and relative computation time (normalized to Tent). DPCore achieves remarkable efficiency by introducing only 0.08M parameters (0.1% of model parameters), whereas ViDA requires 93.70M parameters (7.13M additional parameters, 89× more) and 55.6% more computation time. While this analysis primarily focuses on test-time computation, DPCore also maintains efficiency in pre-adaptation: unlike previous methods which require extensive source data warm-up (e.g., EcoTTA needs 3 epochs of ImageNet training for parameter initialization), DPCore requires minimal preparation, involving only forwarding 300 source examples and computing their statistics. This shows DPCore's efficiency in both the preparation and adaptation phases. Detailed analysis is in Appendix E.3.

**Fixed-Size Coreset Management.** We explore fixed-size coreset strategies for scenarios with strict memory constraints. We evaluate two approaches to maintain a fixed-size coreset on ImageNet-C with K=15: once the number of prompts exceeds K, we either 1) discard the oldest prompt or 2) merge the most similar prompts by averaging those with the smallest statistics distance. As shown in Table 5, both strategies significantly improve upon the source model, with merging performing slightly better than discarding. Since the number of domains is typically unknown in real-world scenarios and the coreset grows only when encountering truly unseen domains, our default approach allows K to evolve naturally.

## 6. Conclusion

We introduce a new CTTA setup: Continual Dynamic Change (CDC) setting that better reflects real-world scenarios where domains recur with varying frequencies and durations. Through extensive benchmarking on four datasets, we demonstrate that previous methods for CTTA struggle in CDC due to convergence issues, catastrophic forgetting, and negative transfer. To remedy this, we propose DPCore, a simple yet effective approach to CTTA that achieves robust performance across diverse domain change patterns while maintaining computational efficiency. DPCore integrates three complementary components: *Visual Prompt Adaptation* for efficient domain alignment with minimal parameters, *Prompt Coreset* for strategic knowledge preservation, and *Dynamic Update* for intelligently managing domain knowledge—updating existing prompts for similar domains while creating new ones for substantially different ones. Through our experiments, we show that DPCore is effective and achieves SOTA performance for classification and segmentation tasks for the standard CSC and the new CDC setting. Our theoretical analysis and comprehensive ablation studies further validate DPCore's effectiveness.

## Acknowledgment

We thank the anonymous reviewers for their insightful comments and suggestions. We are also grateful to Evan Shelhamer for his thoughtful feedback on connecting our work to the broader continual test-time adaptation literature. This work was supported by the NSF EPSCoR-Louisiana Materials Design Alliance (LAMDA) program #OIA-1946231.

## Impact Statement

This paper presents work whose goal is to advance the field of Machine Learning. There are many potential societal consequences of our work, none which we feel must be specifically highlighted here.

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

# DPCore: Dynamic Prompt Coreset for Continual Test-Time Adaptation
## Supplementary Materials

In the Appendix, we provide detailed supplementary materials to enhance understanding of our work: First, we present background materials: related works and CTTA baselines and implementations (Sec. A), and additional details about our method and CDC setting, including hyperparameter selection and theoretical analysis (Sec. B). The experimental aspects are covered in several sections: additional experimental results (Sec. C), discussion of our CDC setting with other CTTA settings (Sec. D), and extended ablation studies (Sec. E). Finally, we discuss limitations of our method in Sec. F.

## A. Related Works and Baselines

### A.1. Continual Learning (CL)

CL (Rebuffi et al., 2017; De Lange et al., 2021; Farajtabar et al., 2020; Rolnick et al., 2019; Li & Hoiem, 2017; Kirkpatrick et al., 2017; Zenke et al., 2017; Silver & Mercer, 2002; Bobu et al., 2018; Mai et al., 2022; 2021; Shim et al., 2021) addresses catastrophic forgetting by enabling models to retain learned information while acquiring new knowledge. This challenge is relevant to to TTA, especially in Continual Dynamic Change (CDC) environments, where adapting to new target domains often degrades performance on previously seen domains. To address these limitations, our approach integrates principles from CL into the TTA framework. We employ a dynamically updating coreset that preserves knowledge across visited domains, alongside a weighted updating mechanism to reduce error accumulation.

### A.2. Baselines

In this section, we provide the details of the CTTA baselines we use in our paper.

**Tent**[1] (**Wang et al., 2021**) updates the Norm Layer parameters through prediction entropy minimization. We follow the same hyperparameters that are set in Tent.

**CoTTA**[2] (**Wang et al., 2022a**) is the first to perform TTA on continually changing domains and propose a teacher-student learning scheme with augmentation-based consistency maximization. We follow the same hyperparameters that are set in CoTTA. The trainable parameters are all the parameters

in ViT-Base.

**DePT** (**Gao et al., 2023**) integrates visual prompts with Vision Transformers to adapt to target domains through memory bank-based pseudo-labeling. This method utilizes a significant number of prompts, which are initialized by warming up on source data.

**SAR**[3] (**Niu et al., 2023**) introduces a sharpness-aware and reliable entropy minimization method. This approach selectively filters noisy samples and optimizes model weights towards stable regions in the parameter space.

**VDP** (**Gan et al., 2023**) employs visual domain prompts in pixel space to adapt to continually changing domains. It dynamically updates lightweight prompts to facilitate domain adaptation, modifying input images instead of model parameters. The pixel prompts are initialized on source data.

**RDumb**[4] (**Press et al., 2023**) employs periodic resets to the original pretrained weights at regular intervals to prevent the accumulation of adaptation errors that lead to performance collapse in continual test-time adaptation.

**EcoTTA**[5] (**Song et al., 2023**) utilizes lightweight meta networks and self-distilled regularization to maintain memory efficiency and ensure long-term adaptation stability. This method emphasizes consistency between the outputs of meta networks and the original frozen network, with meta-networks warmed up on source data for several epochs.

**ViDA**[6] (**Liu et al., 2023b**) uses high-rank and low-rank domain adapters to manage domain-specific and shared knowledge. Designed to address continual test-time adaptation, it dynamically responds to changing domain conditions. The adapters are warmed up on source data.

**BDG** (**Yang et al., 2024**) is a framework designed to balance discriminability and generalization in CTTA by generating high-quality supervisory signals. This framework uses adaptive thresholds for pseudo-label reliability, leverages knowledge from a pre-trained source model to adjust unreliable signals, and calculates a diversity score to ensure future domain generalization.

---

[1] https://github.com/DequanWang/tent
[2] https://github.com/qinenergy/cotta
[3] https://github.com/mr-eggplant/SAR
[4] https://github.com/oripress/CCC
[5] https://github.com/Lily-Le/EcoTTA
[6] https://github.com/Yangsenqiao/vida

**C-MAE**[7] (**Liu et al., 2024**) introduces Adaptive Distribution Masked Autoencoders for CTTA, which employ a Distribution-aware Masking mechanism to adaptively sample masked positions in images.

We utilize official implementations of the method where available; otherwise, we implement it ourselves using the hyperparameters reported in the original paper. To ensure fair comparison, we maintain consistent hyperparameters across both CSC and CDC settings.

## B. Additional Details of DPCore

### B.1. Visual Prompt Adaptation in TTA

In the context of TTA where labels are unavailable, prompts can be optimized through consistency maximization, entropy minimization, or unsupervised distribution alignment (Niu et al., 2024; Sun et al., 2023; Gao et al., 2023; Zhang et al., 2024). We employ distribution alignment due to its proven effectiveness and computational simplicity (Ben-David et al., 2006; Mehra et al., 2024a).

For the current batch $B_t^T$ at time step $t$ from unknown target domain $T$, a prompt is optimized by minimizing the distribution alignment between source and target features:

$$\boldsymbol{p}^* = \arg\min_{\boldsymbol{p}} \ d(\mathcal{Z}^S, \mathcal{Z}_t^T(\boldsymbol{p})),, \tag{8}$$

where $d$ represents a distribution distance and $\mathcal{Z}_t^T(\boldsymbol{p}) = \phi(B_t^T; \theta_\phi, \boldsymbol{p})$ denotes the extracted features of the current batch with prompt $\boldsymbol{p}$, as shown on the right of Fig. 2. We employ marginal distribution alignment (Ben-David et al., 2006; Le et al., 2021; Shen et al., 2018) to avoid potential error accumulation from pseudo labels in continual learning scenarios, though other sophisticated distance metrics are also applicable.

Unlike previous prompt-based TTA methods such as DePT (Gao et al., 2023) or VDP (Gan et al., 2023) that require prompt warm-up on source data for optimal initialization, we initialize prompt tokens from a Gaussian distribution without any warm-up, following Visual Prompt Tuning in supervised settings (Jia et al., 2022). This approach makes our method more practical and efficient.

### B.2. Algorithm of DPCore

To illustrate how DPCore operates as shown in Fig. 2, we present its complete algorithm in Alg. 1.

### B.3. Implementation Details

Here we detail the hyperparameters used in our main experiments. We randomly sample 300 source examples to

---

---

**Algorithm 1** The proposed algorithm DPCore

**Input**: A source pre-trained model $f(x)$, source statistics $\Gamma^S$, test batches $\{B_t\}_{t=1}^T$
**Initialization**: An empty coreset, a pre-defined ratio threshold $\rho$, and random prompt tokens.
1: **for** the first batch $B_1$ **do**
2:     Compute the statistics of batch $B_1$ without prompt.
3:     Learn prompt from scratch (Eq. 8).
4:     Add the batch statistics and prompt to the empty coreset.
5: **end for**
6: **for** $t = 2, ..., T$ **do**
7:     Compute the statistics of batch $B_t$ without prompt.
8:     Compute the weights and weighted prompts (Eq. 6).
9:     Compute batch statistics with the weighted prompt.
10:     Compute ratio of distances with/w.o. weighted prompt.
11:     **if** ratio $\leq \rho$ **then**
12:         Update weighted prompt on $B_t$ by 1 step.
13:         Update all coreset elements (Eq. 7).
14:     **else**
15:         Learn prompt from scratch for $B_t$ (Eq. 8).
16:         Add the batch statistics and learned prompt to the coreset as a new element.
17:     **end if**
18: **end for**
**Output**: Prediction for all batches; The learned coreset.

---

compute the statistics and set the number of prompt tokens to 8 with a test batch size of 64. The updating weight is set as $\alpha = 0.999$. We learn the prompt from scratch for 50 steps and refine the existing prompt for only 1 step. The model is optimized using AdamW with a learning rate of 0.01, and the threshold $\rho$ is set to 0.8. These hyperparameters were determined using four disjoint validation corruptions from ImageNet-C and CIFAR10-C (Hendrycks & Dietterich, 2019): [Speckle Noise, Gaussian Blur, Spatter, Saturate], following (Zhang et al., 2022).

For datasets without validation sets (e.g., CIFAR100-to-CIFAR100C and Cityscapes-to-ACDC), we apply these same hyperparameters without additional tuning to align with practical testing conditions, where hyperparameters must be selected prior to accessing target data. Moreover, we employ identical hyperparameters across both CSC and CDC settings.

### B.4. Comprehensive Analysis of DPCore

We elaborate on the theoretical analysis introduced in Sec. 4.4. For simplicity, we consider a version of Alg. 1 where we use one-hot weights instead of general weights—only the closest core element to a given test batch is considered, and only the mean is stored in the coreset. The simplified algorithm is presented in Alg. 2, denoted as $\mathcal{A}$

Consider batches $B_1, ..., B_t$ naturally clustered into $M$ mutually exclusive clusters $\{G^i\}_{i=1}^M$ based on their distances. Let $\text{Conv}(G^i)$ be the convex hull of batches in

$G^i$, $\text{diam}(S) = \sup_{x,x' \in S} d(x, x')$ be the diameter of set $S$, and $d(S, S') = \inf_{x \in S, x' \in S'} d(x, x')$ be the set distance. We denote by $\Pi^{(t-1)}$ the set of permutations on $(1, ..., t-1)$, by $G_t^i$ the set of batches in cluster $i$ at time $t$, and by $\{\boldsymbol{p}_t^i, \boldsymbol{\mu}_t^i\}$ the $i$-th core element at time $t$. Each core element consists of a core prompt $\boldsymbol{p}_t^i$ and a core mean $\boldsymbol{\mu}_t^i$. Abstractly, the algorithm $\mathcal{A}$ generates the current prompt $\boldsymbol{p}_t = \mathcal{A}(\{B_{t-1}, ..., B_1\})$ using the batch history $\{B_1, ..., B_{t-1}\}$ as input.

**Assumption** (Well-separated clusters): There exists $\theta > 0$ such that:

$$\text{diam}(\text{Conv}(G^i)) < \theta < d(\text{Conv}(G^i), \text{Conv}(G^j)), \ \forall i \neq j.$$

This implies batches within the same cluster are closer to each other than to batches from different clusters.

**Proposition B.1.** *For any $t \geq 1$, Alg. 2 assigns all batches $B_1, ..., B_t$ to correct clusters.*

*Proof.* We prove by induction. At $t = 1$, since no clusters exist, $B_1$ creates the first cluster with $\boldsymbol{\mu}^1 = \boldsymbol{\mu}(B_1)$. Suppose cluster assignments are correct for batches $B_1, ..., B_{t-1}$. Since each core mean $\boldsymbol{\mu}^i$ is updated through interpolation $(1 - \alpha)\boldsymbol{\mu}^i + \alpha\boldsymbol{\mu}(B_t)$, it remains within the convex hull of its cluster: $\boldsymbol{\mu}_t^i \in \text{Conv}(G_t^i)$.

When a new batch $B_t$ belonging to cluster $j$ arrives, we consider four cases based on whether cluster $j$ or other clusters contain batches: 1) If no clusters exist ($|G_{t-1}^j| = |G_{t-1}^k| = 0, \ \forall k \neq j$), $B_t$ creates a new cluster. 2) If only other clusters exist ($|G_{t-1}^j| = 0, \ \text{and} \ \exists k \neq j : |G_{t-1}^k| > 0$), well-separatedness ensures $d(\boldsymbol{\mu}(B_t), \boldsymbol{\mu}^k) > \theta$, so $B_t$ creates a new cluster. 3) If only cluster $j$ exists ($|G_{t-1}^j| > 0, \ \text{and} \ |G_{t-1}^k| = 0, \ \forall k \neq j$), well-separatedness ensures $d(\boldsymbol{\mu}(B_t), \boldsymbol{\mu}^j) < \theta$, so $B_t$ joins cluster $j$. 4) If multiple clusters exist ($|G_{t-1}^j| > 0, \ \text{and} \ \exists k \neq j : |G_{t-1}^k| > 0$), well-separatedness ensures $d(\boldsymbol{\mu}(B_t), \boldsymbol{\mu}^j) < \theta < d(\boldsymbol{\mu}(B_t), \boldsymbol{\mu}^k)$, so $B_t$ correctly joins cluster $j$. □

**Proposition B.2.** *For any $t > 1$, Alg. 2 assigns $B_t$ to the correct cluster independent of batch order $B_{\pi(1)}, ..., B_{\pi(t-1)}, \forall \pi \in \Pi^{(t-1)}$. Furthermore, if $\alpha = \frac{1}{|G^j|}$, then the core mean is the cluster mean: $\boldsymbol{\mu}_t^i = \frac{1}{|G_t^i|} \sum_{B \in G_t^i} \boldsymbol{\mu}(B)$, also independent of batch order.*

*Proof.* From Proposition B.1, correct assignment at time $t$ depends only on well-separatedness and correct assignments at $t - 1$, not on batch order. For the second claim, consider batches $B^{(1)}, B^{(2)}, ...$ are the batches assigned to cluster $j$ in the order they are processed. With $\alpha = \frac{1}{|G^j|}$, updates yield: $\boldsymbol{\mu}^j = \boldsymbol{\mu}(B^{(1)})$, $\boldsymbol{\mu}^j = \frac{1}{2}(\boldsymbol{\mu}(B^{(1)}) + \boldsymbol{\mu}(B^{(2)}))$, $\boldsymbol{\mu}^j = \frac{1}{3}(\boldsymbol{\mu}(B^{(1)}) + \boldsymbol{\mu}(B^{(2)}) + \boldsymbol{\mu}(B^{(3)}))$, and so on. This running average is independent of batch order. □

**Proposition B.3.** *For any $t > 1$, Alg. 2 learns prompt $\boldsymbol{p}_t$ for batch $B_t$ independent of batch order $B_{\pi(1)}, ..., B_{\pi(t-1)}, \forall \pi \in \Pi^{(t-1)}$.*

*Proof.* From Proposition B.2, cluster assignment is independent of batch order. For batch $B_t$, two cases arise: 1) If $B_t$ belongs to an existing cluster $i$, $\boldsymbol{p}_t$ is learned starting from existing core prompt $\boldsymbol{p}^i$. Since $\boldsymbol{p}^i$ is order-independent, $\boldsymbol{p}_t$ is also order-independent. 2) If $B_t$ creates a new cluster, $\boldsymbol{p}_t$ is learned from scratch, making it inherently order-independent. □

Our analysis demonstrates three key properties of DPCore under the well-separatedness assumption: 1) Correct Clustering: DPCore correctly assigns all batches to their respective clusters regardless of sequence length. 2) Order Independence: Cluster assignments remain correct regardless of the order in which batches arrive. 3) Prompt Stability: The learned prompts are independent of batch order.

These properties together show that DPCore can effectively manage domain knowledge even when domains appear in arbitrary orders—a crucial advantage in dynamic environments. While this analysis uses simplified assumptions, it provides valuable insight into DPCore's robustness to random domain changes. The full analysis under general conditions remains an interesting direction for future work.

---

**Algorithm 2** A simplified version of the proposed algorithm

1: **for** $t = 1, 2, ..., T$ **do**
2:     Compute the batch mean $\boldsymbol{\mu}(B_t)$ and distances $d^j = d(\boldsymbol{\mu}(B_t), \boldsymbol{\mu}^j), \ i = 1, ..., K$ against all existing core elements.
3:     **if** coreset is empty or $\min_j d^j > \theta$ **then**
4:         Learn prompt from scratch for $B_t$ (Eq. 8).
5:         Add the batch mean and learned prompt to the coreset as a new element.
6:     **else**
7:         Assign $B_t$ to the closest core element $j^* = \arg\min_j d^j$.
8:         Update the prompt in $j^*$-th core element by 1 step.
9:         Update $j^*$-th core element (Eq. 7).
10:     **end if**
11: **end for**

**Output**: Prediction for all batches; The learned coreset.

---

## C. Additional Results

### C.1. Performance of Single Domain-learned Prompt

To validate the core intuition behind DPCore, we conduct a preliminary experiment examining how a prompt learned from one domain transfers to others. Specifically, we learn a

prompt from scratch on the Gaussian Noise domain and evaluate its effectiveness across the remaining 14 domains. For comparison, we also learn domain-specific prompts independently with known domain boundaries, where the model resets each time it encounters a new domain. As shown in Table 6, the Gaussian Noise-learned prompt yields interesting transfer patterns: it significantly improves performance on similar domains (Shot Noise and Impulse Noise), shows limited effectiveness on others (Defocus Blur and Elastic), and even degrades performance on substantially different domains (Motion and Contrast). These performance variations strongly align with the domain distance measurements illustrated in Fig. 3a, suggesting that the effectiveness of prompt transfer is directly related to the similarity of domains in feature space.

Notably, DPCore surprisingly outperforms the domain-specific prompts despite requiring no domain boundary information. This superior performance stems from DPCore's ability to learn a unified prompt for similar domains (e.g., the entire Noise group including Gaussian, Shot, and Impulse), achieving better adaptation than learning from individual domains in isolation.

These empirical findings reveal two crucial insights that motivate DPCore's design: (1) prompts can transfer effectively between similar domains, potentially enabling more efficient adaptation, and (2) for fundamentally different domains, learning new domain-specific prompts is essential to prevent negative transfer.

## C.2. Results for 10 Random Orders in CSC

This section provides details for the results presented in Fig. 3c. We strictly follow the ten random orders used in CoTTA (Wang et al., 2022a) and evaluate DPCore on ImageNet-to-ImageNet-C across all ten domain orders. It is important to note that even though the domain order is changed, the domain change frequency and domain length remain fixed, resulting in the same domain change pattern in CSC setting. DPCore demonstrates robust performance in this setting, achieving an average error rate of 40.2% with an average coreset size of 13.9 elements. These results highlight the consistency and effectiveness of DPCore across various domain orders, further validating its applicability in real-world scenarios where the order of encountered domains may vary.

## C.3. Results for CIFAR10/100-to-CIFAR10/100C

In this section, we present the results on CIFAR10/100-to-CIFAR10/100C across both CSC and CDC settings. The results for CIFAR10-to-CIFAR10C are in Table 7 and Table 8 for CSC and CDC settings respectively. In the CSC setting, DPCore significantly outperforms all existing methods, achieving a +12.8% improvement over the source pre-

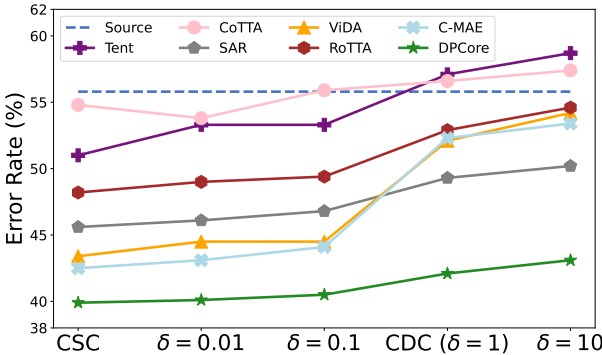

*Figure 6.* Performance comparison across different domain change patterns on ImageNet-to-ImageNet-C. The x-axis shows settings from CSC to CDC with increasing Dirichlet parameter $\delta$, where larger $\delta$ indicates more frequent domain changes. DPCore maintains stable performance even with rapid domain transitions, while other methods degrade significantly.

trained model and surpassing the previous SOTA (ViDA) by 5.3%. This improvement is consistent across all corruption types, with particularly strong performance on noise corruptions (reducing error rates from 60.1% to 22.0% on Gaussian Noise). In the more challenging CDC setting, while previous methods show substantial degradation (ViDA's improvement drops from +7.5% to +4.6%), DPCore maintains robust performance with an 11.1% improvement over the source model.

Similarly, for CIFAR100-to-CIFAR100C task (results in Tables 13 and 14), DPCore demonstrates strong performance across both settings. In CSC, DPCore achieves a 10.3% improvement over the source model and outperforms ViDA by 2.2%, with notable improvements across all corruption types, particularly on Glass (44.1% vs 60.5% source error rate) and Contrast corruptions (13.2% vs 34.8% source error rate). The CDC setting presents a greater challenge, where several methods (CoTTA, VDP) perform worse than the source model, and ViDA's improvement drops from +8.1% to +5.6%. In contrast, DPCore maintains robust performance with an 8.0% improvement over the source model, demonstrating its effectiveness.

## C.4. Comprehensive Results for Cityscapes-to-ACDC

In this section, we provide fine-grained results on Cityscapes-to-ACDC, presented in Table 15 and Table 16 for CSC and CDC settings respectively. We evaluate performance across three rounds and report the average mIoU scores for each domain and round, offering a comprehensive view of adaptation effectiveness over repeated domain sequences.

*Table 6.* **Classification error rate (%) for ImageNet-to-ImageNet-C in the CSC setting.** Gaussian Noise-learned Prompt denotes the prompt learned solely on Gaussian Noise and evaluated across other domains. Domain-specific prompt denote the prompt learned on each domain separately with known domain boundary.

| Algorithm | Gauss. | Shot | Impulse | Defocus | Glass | Motion | Zoom | Snow | Frost | Fog | Bright | Contrast | Elastic | Pixel | JPEG | Mean↓ | Gain↑ |
|---|---|---|---|---|---|---|---|---|---|---|---|---|---|---|---|---|---|
| Source | 53.0 | 51.8 | 52.1 | 68.5 | 78.8 | 58.5 | 63.3 | 49.9 | 54.2 | 57.7 | 26.4 | 91.4 | 57.5 | 38.0 | 36.2 | 55.8 | 0.0 |
| Tent | 52.2 | 48.9 | 49.2 | 65.8 | 73.0 | 54.5 | 58.4 | 44.0 | 47.7 | 50.3 | 23.9 | 72.8 | 55.7 | 34.4 | 33.9 | 51.0 | +4.8 |
| Gaussian Noise-learned Prompt | 42.2 | 40.4 | 42.1 | 67.5 | 65.3 | 60.4 | 66.8 | 50.0 | 41.2 | 56.8 | 27.6 | 95.7 | 56.3 | 37.7 | 32.6 | 52.2 | +3.6 |
| Domain-specific Prompt | 42.2 | 39.9 | 41.6 | 58.8 | 58.1 | 48.6 | 44.0 | 35.0 | 40.6 | 41.5 | 21.1 | 42.9 | 45.6 | 29.6 | 28.0 | 41.2 | +14.6 |
| **DPCore (Proposed)** | **42.2** | **38.7** | **39.3** | **47.2** | **51.4** | **47.7** | **46.9** | **39.3** | **36.9** | **37.4** | **22.0** | **44.4** | **45.1** | **30.9** | **29.6** | **39.9** | **+15.9** |

*Table 7.* **Classification error rate (%) for CIFAR10-to-CIFAR10C in the CSC setting.**

| Algorithm | Gauss. | Shot | Impulse | Defocus | Glass | Motion | Zoom | Snow | Frost | Fog | Bright | Contrast | Elastic | Pixel | JPEG | Mean↓ | Gain↑ |
|---|---|---|---|---|---|---|---|---|---|---|---|---|---|---|---|---|---|
| Source | 60.1 | 53.2 | 38.3 | 19.9 | 35.5 | 22.6 | 18.6 | 12.1 | 12.7 | 22.8 | 5.3 | 49.7 | 23.6 | 24.7 | 23.1 | 28.2 | 0.0 |
| Tent (Wang et al., 2021) | 57.7 | 56.3 | 29.4 | 16.2 | 35.3 | 16.2 | 12.4 | 11.0 | 11.6 | 14.9 | 4.7 | 22.5 | 15.9 | 29.1 | 19.5 | 23.5 | +4.7 |
| CoTTA (Wang et al., 2022a) | 58.7 | 51.3 | 33.0 | 20.1 | 34.8 | 20 | 15.2 | 11.1 | 11.3 | 18.5 | 4.0 | 34.7 | 18.8 | 19.0 | 17.9 | 24.6 | +3.6 |
| VDP(Gan et al., 2023) | 57.5 | 49.5 | 31.7 | 21.3 | 35.1 | 19.6 | 15.1 | 10.8 | 10.3 | 18.1 | 4 | 27.5 | 18.4 | 22.5 | 19.9 | 24.1 | +4.1 |
| ViDA (Liu et al., 2023b) | 52.9 | 47.9 | 19.4 | 11.4 | 31.3 | 13.3 | 7.6 | 7.6 | 9.9 | 12.5 | 3.8 | 26.3 | 14.4 | 33.9 | 18.2 | 20.7 | +7.5 |
| **DPCore(Proposed)** | **22.0** | **18.2** | **14.9** | **14.3** | **24.4** | **13.9** | **12.0** | **11.6** | **10.7** | **15.0** | **5.7** | **21.8** | **15.6** | **12.7** | **18.0** | **15.4** | **+12.8** |

# D. Discussion on CTTA Setting.

## D.1. CDC with Different Dirichlet Distributions

We analyze how different domain change patterns affect CTTA methods by varying the Dirichlet distribution parameter $\delta$ that controls our CDC simulation. As shown in Fig. 6, smaller $\delta$ values (0.01, 0.1) produce domain changes more similar to CSC, while larger values (1, 10) lead to increasingly frequent transitions. DPCore demonstrates remarkable stability across all settings, with error rates increasing only modestly from 39.9% (CSC) to 43.1% ($\delta = 10$), consistently outperforming previous SOTA methods. The stability of other methods, however, deteriorates significantly as $\delta$ increases. In the most challenging scenario ($\delta = 10$) where domains change most rapidly, most previous methods perform similarly to or worse than the source model (55.8%). While SAR maintains some improvement with 50.2% error rate, DPCore significantly outperforms all methods with 43.1%, demonstrating its robust adaptation capability even under extreme domain variation.

## D.2. CDC with Other Distributions

To provide a comprehensive understanding of our CDC setting beyond the Dirichlet distribution, we present a sequence generation method based on two independent distributions: one for domain selection and another for domain duration (number of batches), termed as **CDC-2D**. This two-distribution approach allows us to flexibly control both which domains appear and how long they persist, creating realistic scenarios of domain changes.

Taking uniform distributions as a simple case, we iteratively construct the domain sequence through the following process: at each step, we randomly select a domain from the domain pool using uniform probability, then randomly de-

termine the number of consecutive batches to assign to this domain, also using uniform distribution. If a selected domain has no remaining batches, we perform reselection until finding an available domain. This process continues until all batches across all domains have been assigned, naturally ensuring unpredictable domain changes with varying durations while maintaining complete coverage of all domains.

This uniform distribution-based generation, which we term CDC-2D, offers an alternative perspective on dynamic domain changes compared to the Dirichlet approach in the main paper. As shown in Table 10, we observe two significant trends similar to our main results: First, previous SOTA methods suffer substantial performance degradation when moving from CSC to this CDC-2D setting, with several methods showing minimal improvement over the source model's 55.8% error rate and even the best baseline achieving only moderate gains. This degradation is particularly evident in challenging corruption types where most methods struggle to significantly improve upon the source model. Second, DPCore maintains robust performance with a 43.2% error rate (comparable to 42.1% achieved with Dirichlet distribution) and consistently outperforms other methods, achieving a significant improvement of +12.6% over the source model while the second-best method only achieves +8.1

To further validate DPCore's effectiveness across diverse domain change patterns, we explore CDC settings with different probability distributions. Taking ImageNet-to-ImageNet-C as an example with its 15 corruption domains, we consider probability vectors $[P_1, P_2, ..., P_{15}]$ where each $P_i$ represents the probability of selecting the $i$-th domain. These vectors must satisfy $P_i > 0$ for all $i$ (ensuring every domain has some probability of being selected) and $\sum_{i=1}^{15} P_i = 1$ (making it a valid probability distribution). The uniform

*Table 8.* **Classification error rate (%) for CIFAR10-to-CIFAR10C in the CDC setting.**

| Algorithm | Gauss. | Shot | Impulse | Defocus | Glass | Motion | Zoom | Snow | Frost | Fog | Bright | Contrast | Elastic | Pixel | JPEG | Mean↓ | Gain↑ |
|---|---|---|---|---|---|---|---|---|---|---|---|---|---|---|---|---|---|
| Source | 60.1 | 53.2 | 38.3 | 19.9 | 35.5 | 22.6 | 18.6 | 12.1 | 12.7 | 22.8 | 5.3 | 49.7 | 23.6 | 24.7 | 23.1 | 28.2 | 0.0 |
| Tent (Wang et al., 2021) | 57.4 | 51.7 | 34.2 | 18.2 | 35.1 | 22.4 | 16.9 | 11.8 | 12.6 | 21.6 | 5.1 | 36.7 | 22.1 | 24.3 | 23.2 | 26.2 | +2.0 |
| CoTTA (Wang et al., 2022a) | 59.2 | 54.8 | 45.1 | 20.5 | 36.1 | 21.9 | 19.1 | 11.6 | 12.1 | 21.2 | 6.1 | 35.1 | 22.5 | 21.7 | 19.7 | 27.1 | +1.1 |
| VDP(Gan et al., 2023) | 58.1 | 51.2 | 32.2 | 22.4 | 34.8 | 20.4 | 15.6 | 11.2 | 11.4 | 20.1 | 4.9 | 30.4 | 20.4 | 22.7 | 19.9 | 25.0 | +3.2 |
| ViDA (Liu et al., 2023b) | 56.4 | 50.8 | 28.4 | 16.7 | 32.5 | 15.2 | 13.7 | 10.1 | 10.4 | 13.5 | 4.2 | 26.5 | 19.4 | 35.1 | 21.2 | 23.6 | +4.6 |
| **DPCore(Proposed)** | **24.1** | **20.6** | **24.5** | **13.9** | **26.5** | **14.2** | **13.2** | **13.4** | **10.2** | **12.8** | **5.0** | **22.5** | **16.8** | **20.1** | **18.5** | **17.1** | **+11.1** |

*Table 9.* **Classification error rate (%) for ImageNet-to-ImageNet-C in the CDC-2D setting.**

| Algorithm | Gauss. | Shot | Impulse | Defocus | Glass | Motion | Zoom | Snow | Frost | Fog | Bright | Contrast | Elastic | Pixel | JPEG | Mean↓ | Gain↑ |
|---|---|---|---|---|---|---|---|---|---|---|---|---|---|---|---|---|---|
| Source | 53.0 | 51.8 | 52.1 | 68.5 | 78.8 | 58.5 | 63.3 | 49.9 | 54.2 | 57.7 | 26.4 | 91.4 | 57.5 | 38.0 | 36.2 | 55.8 | 0.0 |
| Tent (Wang et al., 2022a) | 54.1 | 53.2 | 52.9 | 66.7 | 74.6 | 56.6 | 61.2 | 49.2 | 52.3 | 57.5 | 32.1 | 90.4 | 56.1 | 37.7 | 37.1 | 55.4 | +0.4 |
| CoTTA (Wang et al., 2022a) | 52.4 | 52.2 | 51.1 | 68.4 | 76.3 | 57.9 | 62.1 | 48.7 | 53.1 | 55.1 | 26.1 | 88.9 | 57.1 | 37.6 | 38.2 | 55.0 | +0.8 |
| VDP (Gan et al., 2023) | 53.1 | 53.3 | 51.5 | 62.4 | 73.1 | 53.4 | 60.1 | 43.5 | 54.3 | 58.2 | 25.1 | 82.2 | 56.6 | 35.7 | 36.3 | 53.3 | 2.5 |
| SAR (Niu et al., 2023) | 47.0 | 46.1 | 46.1 | 56.1 | 66.4 | 49.9 | 55.1 | 41.1 | 44.8 | 50.9 | 23.6 | 65.3 | 54.2 | 32.9 | 31.4 | 47.4 | +8.1 |
| EcoTTA (Song et al., 2023) | 48.9 | 47.4 | 49.3 | 59.2 | 71.2 | 54.1 | 59.8 | 46.2 | 44.3 | 57.2 | 24.0 | 84.1 | 55.2 | 37.2 | 35.3 | 51.6 | +4.2 |
| ViDA (Liu et al., 2023b) | 48.8 | 49.2 | 47.3 | 56.6 | 71.6 | 55.3 | 59.6 | 41.2 | 49.4 | 60.5 | 27.1 | 83.9 | 57.9 | 34.6 | 34.4 | 51.8 | +4.0 |
| **DPCore (Proposed)** | **42.7** | **40.4** | **42.2** | **57.4** | **61.0** | **51.1** | **52.4** | **35.8** | **41.0** | **38.8** | **22.1** | **55.4** | **48.3** | **31.1** | **28.1** | **43.2** | **+12.6** |

distribution represents a special case where $P_i = \frac{1}{15}$ for all $i$. We randomly generate ten different such probability vectors, each defining a unique pattern of domain changes. The results in Table 10 demonstrate DPCore's remarkable consistency across these varied settings, achieving a mean error rate of 43.5% with standard deviation of 0.7. This stability across different domain change patterns further validates DPCore's robustness and its ability to handle diverse scenarios of distribution shift.

### D.3. Comparison with Other CTTA Settings

In this section, we discuss how our proposed CDC setting differs from other CTTA variants. First, we consider the repeating setting introduced in (Wang et al., 2022a), where 15 corruption domains are repeated for 10 rounds. While this setting eventually accumulates numerous domain changes, it differs fundamentally from CDC: each domain maintains uniform duration (782 batches), and changes occur gradually. This structured nature makes it more similar to CSC than our CDC setting, where domain durations vary significantly and changes occur more frequently.

Second, the gradual domain change setting proposed in (Lee et al., 2024) assumes blurred domain boundaries, where batches near transitions contain mixed-domain data. While this realistic assumption differs from traditional CTTA methods, it still follows the CSC pattern where mixed batches constitute only a small portion of all data due to long domain durations. Although we haven't directly evaluated DPCore in this setting, our proposed variant DPCore-B (detailed in Section E.1) demonstrates effective handling of mixed-domain batches, suggesting potential applicability to gradual domain changes.

Third, the Practical Test-Time Adaptation (PTTA) setting

(Yuan et al., 2023) assumes label-balanced batches with local class correlation (e.g., single-class batches) in CSC. This contrasts with our setting, which follows the common CTTA assumption (Wang et al., 2022a; Niu et al., 2022; Liu et al., 2023b; 2024) of uniform sampling and label balance. Interestingly, despite not being designed for PTTA, DPCore shows substantial improvement in this setting (results in Table 11). Meanwhile, RoTTA (Yuan et al., 2023), though specifically designed for PTTA, demonstrates limited effectiveness on ImageNet-C with ViTs compared to its stronger performance on smaller datasets like CIFAR10, consistent with findings in (Marsden et al., 2023). Furthermore, RoTTA's performance degrades significantly in our CDC setting. In contrast, DPCore maintains robust performance across CSC, CDC, and PTTA settings, demonstrating its versatility across different test-time scenarios.

Our CDC setting introduces unique challenges through its combination of frequent domain changes and varying domain durations, distinguishing it from previous CTTA variants. This more dynamic and unpredictable environment better reflects real-world scenarios while posing significant challenges for existing CTTA methods.

### E. Additional Ablation Studies

### E.1. DPCore with Small Batch Size

While DPCore performs effectively with batch sizes of four or larger (as shown in the Fig. 5d), significant challenges emerge with extremely small batches, particularly in single-sample scenarios. To address this limitation, we propose DPCore-B (DPCore with Buffer Zone), which accumulates a predefined number of samples ($D$) in the buffer zone before performing any updates. During this accumulation period, we temporarily store the `[CLS]` features from each

*Table 10.* **Average error rate (%) of DPCore across 10 different CDC-2D settings (R1-10) on ImageNet-to-ImageNet-C.**

|  | **Source** | Uniform | R1 | R2 | R3 | R4 | R5 | R6 | R7 | R8 | R9 | R10 | **Average** | **Std** |
|---|---|---|---|---|---|---|---|---|---|---|---|---|---|---|
| Err Mean↓ | 55.8 | 43.2 | 42.1 | 42.8 | 44.2 | 43.5 | 43.5 | 42.7 | 43.9 | 44.4 | 44.7 | 43.0 | 43.5 | 0.7 |

*Table 11.* Average error rate (%) of DPCore and RoTTA across various CTTA settings on ImageNet-to-ImageNet-C. The improvement over the source model is shown in parentheses.

| Algorithm | CSC | PTTA | CDC | Average |
|---|---|---|---|---|
| **Source** | 55.8 | 55.8 | 55.8 | 0 |
| **RoTTA** | 48.2 (+7.6) | 49.5 (+6.35) | 53.1 (+2.7) | 50.3 (+5.6) |
| **DPCore** | **39.9 (+15.9)** | **43.9 (+12.0)** | **42.1 (13.7)** | **42.0 (+13.9)** |

*Table 12.* Average error rate (%) of DPCore-B across various buffer zone size $D$ on ImageNet-to-ImageNet-C in CSC setting when batch size is one.

|  | **Source** | D= 8 | D= 16 | D= 32 | D= 64 |
|---|---|---|---|---|---|
| **Err Mean↓** | 55.8 | 43.0 | 42.1 | 41.6 | 41.2 |

batch.

Our evaluation of DPCore-B in the CSC setting with various values of $D$ shows promising results (Table 12), effectively maintaining DPCore's performance even with small batches. However, this success is limited to the CSC setting where domain changes are structured and predictable. Even when accumulated samples span domain boundaries (violating the single-domain batch assumption shared by most CTTA methods, including ours), the impact remains minimal in CSC due to the limited number of domain transitions. For instance, on ImageNet-to-ImageNet-C with batch size 64, each domain has 782 batches, totaling 11,730 batches across 15 domains. Only 14 transition points potentially contain mixed-domain data, causing a negligible impact on overall performance.

In contrast, CDC presents a fundamentally more challenging scenario where domain transitions occur frequently and unpredictably. The substantially higher number of potential mixed-domain batches in CDC violates our method's assumptions more severely, making DPCore-B ineffective. This limitation further demonstrates why CDC, with its frequent domain changes and varying durations, better reflects real-world challenges compared to conventional CSC settings. The development of effective small-batch adaptation strategies for CDC remains an important direction for future research.

**E.2. DPCore without Access to Source Data**

In this section, we explore an extreme scenario where source data is completely inaccessible, further demonstrating DP-

Core's practicality in real-world applications. Following the insights from (Mehra et al., 2024a), we propose using public datasets that share similar or identical label spaces. For instance, in ImageNet-to-ImageNet-C or CIFAR10-to-CIFAR10C tasks, we leverage STL10 (Coates et al., 2011), which shares label space with CIFAR10 despite being distinct from ImageNet.

To ensure quality reference data, we filter the public dataset using the source model with an entropy-based threshold. Specifically, following (Niu et al., 2022), we set the threshold as $0.4 \times \ln C$, where $C$ is the number of classes in the task (e.g., 1000 for ImageNet-to-ImageNet-C). We maintain consistency with our main experiments by selecting 300 unlabeled samples from STL10 using this filtering strategy and compute statistics to replace the source statistics. Remarkably, as shown in Fig. 5c, using these 300 filtered STL10 images as reference data for ImageNet-C experiments still achieves an error rate of 45.6%, yielding a substantial 10.2% improvement over the source model. These results demonstrate that DPCore can maintain effective adaptation even without access to source data, further establishing its practical utility in real-world scenarios where source data might be unavailable due to privacy, security, or storage constraints.

**E.3. Details in Computational Complexity**

We provide a detailed analysis of the computation and memory efficiency of DPCore. In Table 4, we list the total number of trainable parameters required for ImageNet-to-ImageNet-C tasks. The parameter count for DPCore is not static; it increases with the addition of more core elements. The number of core elements depends on the number of unseen domains encountered. However, we demonstrate in Fig. 3c that the core set size remains stable across different domain orders of 15 corruptions in the CSC setting.

For ImageNet-to-ImageNet-C tasks, the total number of parameters is computed as the number of core elements (14) multiplied by the prompt length (8) and the dimension of each prompt token (768). The counts for forward and backward propagations are averaged per batch. In DPCore, a new prompt is learned from scratch over 50 steps only when a test batch is evaluated as originating from a potential new domain. This results in the forward and backward operations being repeated for 50 steps for these batches. Notably, only 14 prompts undergo this extensive learning process out of 11,730 batches. For the remaining batches, DPCore updates the weighted prompt in a sin-

gle step (one forward and one backward operation) and then uses the updated weighted prompt to refine all existing prompts without additional forward or backward passes. Therefore, the average number of backward propagations per batch is $\frac{14 \times 50 + (11730-14) \times 1}{11730} \approx 1.06$. For forward propagation, each batch requires two additional forward passes (one without prompt, one with weighted prompt) for evaluation, leading to an average of $1.06 + 2 = 3.06$ forward passes.

The computational overhead of learning new prompts becomes negligible given the large number of total batches. Additionally, the parameters introduced by the prompts are minimal compared to the parameter increases required by other CTTA methods. Consequently, our prompt parameters do not require any warm-up, which conserves memory and reduces computation both before and during adaptation.

# F. Discussions and Limitations

### F.1. Potential Failure Cases

Our method, DPCore, is designed to be robust against scenarios such as *Overlapping Distributions*, *Noisy or Small Domain Shifts*, and *Inconsistent Weighting Effects* being inherent failure cases. The core objective of DPCore is not to identify an identical number of prompts as domains or to pinpoint every distinct domain. Instead, it aims to update the same prompt for groups of similar domains. This is supported by our findings (Sec. 4.3 and Appendix C.1) that a prompt learned for one domain (e.g., Gaussian Noise) can be effective for a similar domain (e.g., Shot Noise) and can be further improved with minor updates. Due to the use of distance-induced weights, a new domain can be effectively represented as a decomposition of existing domains, allowing the weighted prompt to perform well even if derived from different domains, as domain similarity is assessed by distance and prompts are learned through distance minimization. A more significant potential failure case arises when each test batch contains data from multiple domains. In this situation, batch statistics would become unstable, potentially leading DPCore to treat each batch as a new, distinct domain. This would significantly reduce efficiency, as the algorithm would need to learn prompts from scratch for each batch, a scenario analogous to our experimental setup in Table 3 Exp-2 (where prompts were learned from scratch for each batch, though each batch was from a single domain). Such multi-domain batches pose a challenge to most CTTA methods, which typically assume single-domain test batches. We identify this as an avenue for future investigation.

### F.2. Limitations

While our method significantly advances CTTA, it has certain limitations that may affect its broader application. Firstly, DPCore requires access to source statistics or reference data, which may not always be available, especially in scenarios where the source data is proprietary or sensitive. Secondly, DPCore assumes that each batch of data comes from the same domain, which may not hold true in mixed-domain environments. In real-world applications, data streams could contain a mix of samples from various domains, and the method's performance in such scenarios remains to be investigated. Furthermore, while DPCore demonstrates impressive efficiency compared to other methods, it still introduces additional computational overhead during the adaptation phase, which may pose challenges in resource-constrained environments or applications with strict latency requirements.

Table 13. **Classification error rate (%) for CIFAR100-to-CIFAR100C in the CSC setting.**

| Algorithm | Gauss. | Shot | Impulse | Defocus | Glass | Motion | Zoom | Snow | Frost | Fog | Bright | Contrast | Elastic | Pixel | JPEG | Mean↓ | Gain↑ |
|---|---|---|---|---|---|---|---|---|---|---|---|---|---|---|---|---|---|
| Source | 55.0 | 51.5 | 26.9 | 24.0 | 60.5 | 29.0 | 21.4 | 21.1 | 25.0 | 35.2 | 11.8 | 34.8 | 43.2 | 56.0 | 35.9 | 35.4 | 0.0 |
| Tent (Wang et al., 2021) | 53.0 | 47.0 | 24.6 | 22.3 | 58.5 | 26.5 | 19.0 | 21.0 | 23.0 | 30.1 | 11.8 | 25.2 | 39.0 | 47.1 | 33.3 | 32.1 | +3.3 |
| CoTTA (Wang et al., 2022a) | 55.0 | 51.3 | 25.8 | 24.1 | 59.2 | 28.9 | 21.4 | 21.0 | 24.7 | 34.9 | 11.7 | 31.7 | 40.4 | 55.7 | 35.6 | 34.8 | +0.6 |
| VDP (Gan et al., 2023) | 54.8 | 51.2 | 25.6 | 24.2 | 59.1 | 28.8 | 21.2 | 20.5 | 23.3 | 33.8 | 7.5 | 11.7 | 32.0 | 51.7 | 35.2 | 32.0 | +3.4 |
| ViDA (Liu et al., 2023b) | 50.1 | 40.7 | 22.0 | 21.2 | 45.2 | 21.6 | 16.5 | 17.9 | 16.6 | 25.6 | 11.5 | 29.0 | 29.6 | 34.7 | 27.1 | 27.3 | +8.1 |
| **DPCore (Proposed)** | **48.2** | **40.2** | **21.3** | **20.2** | **44.1** | **21.1** | **16.2** | **18.1** | **15.2** | **22.3** | **9.4** | **13.2** | **28.6** | **32.8** | **25.5** | **25.1** | **+10.3** |

Table 14. **Classification error rate (%) for CIFAR100-to-CIFAR100C in the CDC setting.**

| Algorithm | Gauss. | Shot | Impulse | Defocus | Glass | Motion | Zoom | Snow | Frost | Fog | Bright | Contrast | Elastic | Pixel | JPEG | Mean↓ | Gain↑ |
|---|---|---|---|---|---|---|---|---|---|---|---|---|---|---|---|---|---|
| Source | 55.0 | 51.5 | 26.9 | 24.0 | 60.5 | 29.0 | 21.4 | 21.1 | 25.0 | 35.2 | 11.8 | 34.8 | 43.2 | 56.0 | 35.9 | 35.4 | 0.0 |
| Tent (Wang et al., 2021) | 53.5 | 46.5 | 26.2 | 25.8 | 61.0 | 28.1 | 23.5 | 21.7 | 22.6 | 30.9 | 11.0 | 24.4 | 41.3 | 49.1 | 36.7 | 33.5 | +1.9 |
| CoTTA (Wang et al., 2022a) | 53.3 | 51.9 | 27.1 | 26.3 | 60.5 | 29.2 | 21.3 | 22.5 | 23.5 | 35.6 | 13.7 | 33.7 | 41.6 | 59.1 | 39.4 | 35.9 | -0.5 |
| VDP (Gan et al., 2023) | 56.6 | 53.5 | 31.8 | 29.1 | 63.9 | 33.9 | 23.5 | 25.7 | 29.9 | 38.5 | 12.1 | 15.5 | 34.0 | 53.8 | 39.6 | 36.1 | -0.7 |
| ViDA (Liu et al., 2023b) | 50.1 | 42.1 | 23.9 | 23.3 | 48.1 | 23.7 | 19.5 | 18.7 | 18.5 | 29.6 | 11.6 | 36.1 | 32.6 | 37.1 | 32.8 | 29.8 | +5.6 |
| **DPCore (Proposed)** | **54.0** | **42.5** | **23.5** | **22.8** | **45.3** | **21.4** | **18.6** | **21.2** | **16.8** | **23.2** | **10.0** | **15.1** | **35.7** | **34.6** | **25.9** | **27.4** | **+8.0** |

Table 15. **mIoU score for Cityscapes-to-ACDC in the CSC setting.** The same target domains are repeated three times.

| Algorithm | Round 1 | | | | | Round 2 | | | | | Round 3 | | | | | Mean↑ | Gain↑ |
|---|---|---|---|---|---|---|---|---|---|---|---|---|---|---|---|---|---|
| | Fog | Night | Rain | Snow | Mean↑ | Fog | Night | Rain | Snow | Mean↑ | Fog | Night | Rain | Snow | Mean↑ | | |
| Source | 69.1 | 40.3 | 59.7 | 57.8 | 56.7 | 69.1 | 40.3 | 59.7 | 57.8 | 56.7 | 69.1 | 40.3 | 59.7 | 57.8 | 56.7 | 56.7 | 0.0 |
| Tent (Wang et al., 2021) | 69.0 | 40.2 | 60.1 | 57.3 | 56.7 | 68.3 | 39.0 | 60.1 | 56.3 | 55.9 | 67.5 | 37.8 | 59.6 | 55.0 | 55.0 | 55.7 | -1.0 |
| CoTTA (Wang et al., 2022a) | 70.9 | 41.2 | 62.4 | 59.7 | 58.6 | 70.9 | 41.1 | 62.6 | 59.7 | 58.6 | 70.9 | 41.0 | 62.7 | 59.7 | 58.6 | 58.6 | +1.9 |
| DePT (Gao et al., 2022) | 71.0 | 40.8 | 58.2 | 56.8 | 56.5 | 68.2 | 40.0 | 55.4 | 53.7 | 54.3 | 66.4 | 38.0 | 47.3 | 47.2 | 49.7 | 53.4 | -3.3 |
| VDP (Gan et al., 2023) | 70.5 | 41.1 | 62.1 | 59.5 | 58.3 | 70.4 | 41.1 | 62.2 | 59.4 | 58.2 | 70.4 | 41.0 | 62.2 | 59.4 | 58.2 | 58.2 | +1.5 |
| SAR (Niu et al., 2023) | 69.0 | 40.2 | 60.1 | 57.3 | 56.7 | 69.0 | 40.3 | 60.0 | 67.8 | 59.3 | 67.5 | 37.8 | 59.6 | 55.0 | 55.0 | 57.0 | +0.3 |
| EcoTTA (Song et al., 2023) | 68.5 | 35.8 | 62.1 | 57.4 | 56.0 | 68.3 | 35.5 | 62.3 | 57.4 | 55.9 | 68.1 | 35.3 | 62.3 | 57.3 | 55.8 | 55.8 | -0.9 |
| ViDA (Liu et al., 2023b) | 71.6 | 43.2 | 66.0 | 63.4 | 61.1 | 73.2 | 44.5 | 67.0 | 63.9 | 62.2 | 73.2 | 44.6 | 67.2 | 64.2 | 62.3 | 61.9 | +5.2 |
| C-MAE (Liu et al., 2024) | 71.9 | 44.6 | 67.4 | 63.2 | 61.8 | 71.7 | 44.9 | 66.5 | 63.1 | 61.6 | 72.3 | 45.4 | 67.1 | 63.1 | 62.0 | 61.8 | +5.1 |
| **DPCore (Proposed)** | **71.7** | **47.2** | **66.1** | **63.3** | **62.1** | **73.0** | **47.8** | **66.5** | **63.1** | **62.6** | **73.4** | **47.8** | **67.1** | **62.7** | **62.8** | **62.5** | **+5.8** |

Table 16. **mIoU score for Cityscapes-to-ACDC in the CDC setting.** The same target domains are repeated three times.

| Algorithm | Round 1 | | | | | Round 2 | | | | | Round 3 | | | | | Mean↑ | Gain↑ |
|---|---|---|---|---|---|---|---|---|---|---|---|---|---|---|---|---|---|
| | Fog | Night | Rain | Snow | Mean↑ | Fog | Night | Rain | Snow | Mean↑ | Fog | Night | Rain | Snow | Mean↑ | | |
| Source | 69.1 | 40.3 | 59.7 | 57.8 | 56.7 | 69.1 | 40.3 | 59.7 | 57.8 | 56.7 | 69.1 | 40.3 | 59.7 | 57.8 | 56.7 | 56.7 | 0.0 |
| Tent (Wang et al., 2021) | 68.5 | 39.8 | 59.7 | 58.3 | 56.6 | 67.0 | 38.1 | 58.9 | 55.3 | 54.8 | 66.1 | 36.4 | 58.4 | 52.6 | 53.4 | 54.9 | -1.8 |
| CoTTA (Wang et al., 2022a) | 70.1 | 41.2 | 61.9 | 58.6 | 58.0 | 69.7 | 41.0 | 61.4 | 57.2 | 57.3 | 68.7 | 40.5 | 60.3 | 56.4 | 56.5 | 57.3 | +0.5 |
| DePT (Gan et al., 2023) | 69.5 | 40.2 | 58.7 | 56.2 | 56.2 | 68.4 | 39.8 | 58.3 | 55.7 | 55.6 | 66.7 | 39.1 | 57.9 | 54.3 | 54.5 | 55.4 | -1.3 |
| VDP (Gan et al., 2023) | 70.5 | 40.8 | 61.9 | 59.2 | 58.1 | 70.1 | 40.3 | 61.2 | 57.8 | 57.4 | 70.0 | 39.2 | 59.9 | 57.6 | 56.7 | 57.4 | +0.7 |
| SAR (Niu et al., 2023) | 69.1 | 41.6 | 59.9 | 57.5 | 57.0 | 68.8 | 41.2 | 59.7 | 57.4 | 56.8 | 69.1 | 40.8 | 59.2 | 57.0 | 56.5 | 56.8 | +0.1 |
| EcoTTA (Song et al., 2023) | 68.4 | 34.6 | 61.8 | 57.4 | 55.6 | 68.1 | 34.3 | 61.6 | 57.2 | 55.3 | 68.1 | 33.4 | 61.7 | 57.0 | 55.1 | 55.3 | -1.4 |
| ViDA (Liu et al., 2023b) | 71.0 | 42.1 | 64.2 | 62.9 | 60.1 | 70.6 | 41.5 | 62.1 | 62.1 | 59.1 | 70.2 | 40.9 | 61.5 | 61.6 | 58.6 | 59.2 | +2.5 |
| **DPCore (Proposed)** | **71.9** | **46.3** | **64.1** | **63.3** | **61.4** | **71.6** | **45.1** | **63.4** | **63.1** | **60.8** | **71.6** | **44.2** | **63.9** | **63.2** | **60.7** | **61.0** | **+4.3** |

