# OpenReview forum: "DPCore: Dynamic Prompt Coreset for Continual Test-Time Adaptation"
_ICML.cc/2025/Conference — ICML 2025 poster_

### Official Review · Reviewer_jKst · 2025-03-13

**Overall Recommendation:** 2

**Summary:**

This paper proposes DPCore for Continual Test-Time Adaptation. It integrates a Visual Prompt Adaptation for efficient domain alignment, a Prompt Coreset for knowledge preservation, and a Dynamic Update mechanism. Extensive experiments on four benchmarks demonstrate that DPCore outperforms existing CTTA methods.

**Claims And Evidence:**

Yes

**Essential References Not Discussed:**

no

**Experimental Designs Or Analyses:**

Yes

**Methods And Evaluation Criteria:**

The proposed DPCore presents a novel solution for the Continual Dynamic Change (CDC) setup.

**Other Comments Or Suggestions:**

no

**Other Strengths And Weaknesses:**

Strengths:
1. The idea that addressing Continual Dynamic Change (CDC) setup is interesting;
2. Using Online K-Means to generate Prompt Coreset is reasonable.

Weaknesses:
1. The visual prompt adaptation is not a novel solution for domain alignment;
2. This paper requires extra hyperparameters, e.g. K, the temperature \tau and \alpha, while the authors do not clarify their settings.
3. The updating mechanism is not very persuasive, as the linear combination of the coreset element in Eq. 7 can not depict the domain gaps.

**Questions For Authors:**

1. Theoretically, the hyperparameter K, the temperature \tau and \alpha will affect the final results, why don’t you provide detailed ablation analyses?
2. What’s the major contribution of the visual prompt adaptation?

**Relation To Broader Scientific Literature:**

The Continual Dynamic Change (CDC) setup is novel and interesting, this paper explores to address this new problem in a memory efficient way.

**Theoretical Claims:**

The Comprehensive Analysis of DPCore in section C.4 has been checked.

---

> ### Author Rebuttal · Authors · 2025-03-31
>
> We appreciate the reviewer's feedback and address the specific concerns raised:
> ## Q1. Hyperparameter Analysis
> We determine hyperparameters using four disjoint validation corruptions from ImageNet-C and CIFAR10-C. The same hyperparameters (detailed in Sec 4.1, Appendix C.3) are used across all experiments. Additional results on ImageNet-C:
> ### 1. Impact of Temperature τ
> The temperature τ in Eq.6 controls weight assignment softness. We evaluated τ in [0.1, 10.0]:
> |τ|0.1|0.3|0.5|1.0|3.0|5.0|10.0|
> |-|-|-|-|-|-|-|-|
> |Err Rate(%)|45.1|44.7|42.9|40.2|39.9|40.3|43.5|
>
> DPCore is stable for τ between 1.0 and 5.0. We used τ=3.0 for all experiments.
> ### 2. Impact of Update Weight α
> We use exponential moving average in updating mechanism (Eq.7). We didn't tune α but set α=0.999 reported in [1] (comparable to ViDA: 0.999, CoTTA: 0.99). We evaluated α in [0.7, 0.999] and found stable performance when α≥0.9:
> |α|0.7|0.8|0.9|0.95|0.99|0.999|
> |-|-|-|-|-|-|-|
> |Err Rate(%)|42.1|41.3|40.2|40.0|39.6|39.9|
> ### 3. Impact of Coreset Size K
> The parameter K is not fixed but grows dynamically as new domains are encountered, better aligning with real-world scenarios where the number of unseen domains is usually **unknown**. It is not a hyperparameter that needs to be specified but rather an outcome of the algorithm. More results and discussion can be found in Fig.3b and our response to Reviewer te3N Q4.
> ## Q2. Contribution of Visual Prompt Adaptation (VPA)
> We agree that VPA itself is not novel, as noted in our paper with citations. Our key contribution is not the VPA component in isolation but how it is integrated within the **dynamic coreset** framework:
> 1. **VPA as a Practical Component**: While not new, VPA offers a lightweight, efficient adaptation mechanism well-suited for our dynamic coreset approach. As shown in Table 3 (Exp-1), using VPA alone achieves only +5.0% improvement, but when combined with our dynamic coreset, it achieves +15.9%.
> 2. **VPA is Replaceable**: Table 3 (Exp-3) shows our dynamic coreset approach isn't dependent on VPA. Replacing VPA with NormLayer parameters still achieves strong performance (+10.7%), highlighting that our main contribution is the dynamic coreset approach rather than VPA.
> 3. **Efficient Domain Alignment**: Our VPA implementation offers computational advantages in changing environments. It requires minimal parameters (0.1% of model parameters) and few source examples (300 unlabeled), making it practical for real-world deployment.
>
> In summary, while VPA itself isn't our core contribution, its integration in our dynamic coreset framework is key to DPCore's effectiveness. Our novelty lies in how we manage domain knowledge through the dynamic coreset, not in the specific adaptation method (e.g., VPA).
> ## Q3. Updating Mechanism Concerns
> Our updating mechanism is novel and effective for several reasons:
> 1. It works in tandem with our objective function (Eq.5), which creates a meaningful mapping between feature statistics and prompts, effectively reflecting the domain gaps. Empirical results support this: Table 5 shows that prompts learned for similar domains (with similar distances) transfer effectively between them; Fig.3a shows our prompts consistently reduce domain gaps; Fig.3c confirms stability across diverse domain orders.
> 2. The mechanism only activates when the current batch belongs to a seen domain or is similar to seen domains. For completely new domains (with large domain gaps), it is not used. Instead, existing core elements stay unchanged while a new prompt is learned from scratch and added to the coreset to prevent negative transfer.
> 3. Evaluating each prompt separately and selecting the closest one poses challenges (Sec 3.3): 1) It processes same test batch multiple times (linear to the coreset size), increasing computation. Linear combination reduces computation to constant time (only once); 2) Nearest neighbor selection might diminish coreset power, particularly for unseen domains which could be viewed as decompositions of known domains; 3) Increasing temperature τ to prioritize the nearest neighbor does yield heavier weighting, but performance drops (see Q1.1 τ table).
> 4. The linear combination used in Eq.7, while simplifying domain relationships, is highly effective. Table 1 and Fig.4b show DPCore consistently outperforms more complex methods across both CSC and CDC settings. We chose this approach for its efficiency in handling not only visual prompts but also other parameters (e.g., NormLayer parameters from Q2.2), balancing effectiveness and simplicity, and achieving SOTA performance with computational efficiency.
>
> We appreciate the reviewer's feedback on our work. We believe DPCore makes a significant contribution to continual test-time adaptation, particularly for the challenging CDC setting that better reflects real-world scenarios.
>
> [1] "Mean teachers are better role models: Weight-averaged consistency targets improve semi-supervised deep learning results" NeurIPS2017

---

> > ### Comment · Reviewer_jKst · 2025-04-08
> >
> > I notice that my first two concerns have been addressed in the rebuttal. However, for the updating mechanism, the authors have not provided convincing analyses to illustrate how the updating leads to more powerful model adaptation. Besides, the results may be sensitive to the predefined threshold ratio $\rho$ and $\alpha$.

---

> > > ### Author Response · Authors · 2025-04-08
> > >
> > > Dear Reviewer jKst,
> > >
> > > Thank you for acknowledging our response to your first two concerns. We appreciate your feedback and would like to provide additional clarification:
> > >
> > > **Summary**: Our updating mechanism is a critical component that improves performance by 8.4% while reducing computation by 6x. Its effectiveness is thoroughly demonstrated in our experiments, and it shows stable performance across a wide range of hyperparameter values (α and ρ), making it robust for practical applications.
> > >
> > > ## C1. The importance of updating mechanism
> > > Our updating mechanism addresses key challenges identified in Sec 3.3 "Motivations" (Lines 183-197) and Appendix D.1, where we observed that:
> > >
> > > 1. Prompts learned on one domain (e.g., Gaussian Noise) can work effectively on similar domains (e.g., Shot Noise).
> > > 2. Performance improves by dynamically updating existing prompts on similar domains.
> > > 3. However, prompts can be ineffective or harmful for substantially different domains (increasing error rate from 91.4% to 95.7% on Contrast).
> > >
> > > The effectiveness of our updating mechanism is demonstrated through multiple analyses:
> > >
> > > 1. As shown in Table 3 Exp-2, without dynamic updating (DU), performance improvement drops significantly (+7.5% vs. +15.9% with DU) and computation time increases 6x compared to DPCore.
> > > 2. Fig.3a shows our approach with updating mechanism (green curve) consistently reduces domain gaps across all corruption types more effectively than static prompts (pink curve).
> > > 3. Table 5 demonstrates our approach achieves higher improvement (+15.9%) than static prompts (+3.6%).
> > >
> > > Our updating mechanism is particularly powerful in the CDC setting by:
> > > 1. Learning new prompts from scratch while keeping existing ones fixed when domains differ substantially (more frequently changing domains).
> > > 2. Updating the same prompts across similar domains even when they don't appear continuously (brief domain length).
> > >
> > > This explains the superior performance of our method in CDC settings.
> > > ## C2. The sensitivity to updating weight α and threshold ratio ρ
> > > For updating weight α:
> > > 1. We adopt α=0.999 from prior work [1], used in CoTTA and ViDA.
> > > 2. Our method is stable for α ≥ 0.9, commonly used values for Exponential Moving Average (EMA).
> > > 3. Even with smaller ρ values (0.7, 0.8), error rates (42.1%, 41.3%) significantly outperform baselines (CoTTA: 54.8%, ViDA: 43.4%).
> > >
> > > For threshold ratio ρ:
> > >
> > > 1. All hyperparameters are determined using disjoint validation corruptions (Lines 307-310) **before** accessing to the test data.
> > > 2. We use ρ=0.8 consistently across all datasets and settings.
> > > 3. Additional sensitivity analysis in [figure](https://anonymous.4open.science/r/DPCore-Supp-8D17/ablation_rho_all.png) shows stable performance across all three datasets (ImageNet-C, CIFAR10-C, and CIFAR100-C). For example, on ImageNet-C:
> > >
> > >     - ρ ∈ [0.6, 0.9]: 39.8±0.2% error rate.
> > >     - ρ ∈ [0.4, 1.0]: 40.6±1.0% error rate.
> > >     - Even at ρ=0.1, our method (48.3%) still improves the source model by +7.5% and outperforms SOTA methods like CoTTA (54.8%).
> > >
> > > We hope these clarifications address your concerns and kindly request you to reconsider the rating/scoring. We're happy to provide any additional details if needed.
> > >
> > > [1] "Mean teachers are better role models: Weight-averaged consistency targets improve semi-supervised deep learning results" NeurIPS2017
> > >
> > > Best regards,
> > >
> > > Authors

---

### Official Review · Reviewer_te3N · 2025-03-13

**Overall Recommendation:** 4

**Summary:**

This paper introduces DPCore, a novel approach to Continual Test-Time Adaptation (CTTA) that addresses challenges in dynamically changing environments where domains recur with varying frequencies and durations. DPCore employs Visual Prompt Adaptation for efficient domain alignment, a Prompt Coreset for knowledge retention, and a Dynamic Update Mechanism to intelligently adjust or create prompts based on domain similarity. Extensive experiments on four benchmarks show that DPCore achieves state-of-the-art.

**Claims And Evidence:**

The authors mention that storing prompts and statistics is more memory-efficient (Lines 216–219). However, the authors do not provide a quantitative comparison. Can the authors present a memory consumption analysis, specifically comparing the storage requirements of their approach to storing coreset representations? This would help validate the claimed efficiency advantage.

**Essential References Not Discussed:**

The related works are adequately discussed.

**Experimental Designs Or Analyses:**

The evaluation of the proposed method is comprehensive, and the authors have conducted extensive empirical studies to validate its effectiveness.

**Methods And Evaluation Criteria:**

1. The proposed new setting is practical and aligns well with real-world dynamic changes. The authors have also conducted studies to demonstrate its relevance and value.
2.  The proposed method is novel and effectively addresses the challenges posed by the new setting.
3.  The benchmark used in this paper is appropriate and provides a meaningful evaluation of the proposed method's performance.

**Other Comments Or Suggestions:**

No additional comments

**Other Strengths And Weaknesses:**

The paper is well-written, and the proposed new setting is valuable. The method is novel and demonstrates strong performance in this setting. The authors have conducted extensive analysis and experiments, making the empirical results highly compelling.

Weaknesses:

1.	Sensitivity of the Hyperparameter \rho: The new prompt is trained when a batch of data is identified as belonging to a new domain. However, the hyperparameter \rho may be sensitive to the dataset. Can the authors provide a sensitivity analysis of  \rho across different datasets and generate figures similar to Figure 5(b) to illustrate its impact?
2.	Memory Overhead of Storing Prompt Coresets: The proposed method requires retaining a prompt coreset, and as the number of domains increases, the coreset size may grow linearly. To improve memory efficiency, can the authors explore merging similar domain prompts to reduce storage requirements? An alternative approach could be maintaining a fixed-size coreset while ensuring that it remains representative, which would better align with practical memory constraints.

**Questions For Authors:**

The authors use the distance between prompts with and without weighting to identify new domains. However, it is important to understand under what circumstances this method might fail to correctly identify a new domain.

Can the authors explain the potential failure cases? Specifically, are there scenarios where:

1.	Overlapping Distributions: The prompt distances for new and existing domains are too similar, making it difficult to distinguish between them?
2.	Noisy or Small Domain Shifts: If the domain shift is subtle or gradual, could the method fail to detect a new domain?
3.	Inconsistent Weighting Effects: Are there cases where the weighting mechanism leads to misleading prompt distances, potentially misclassifying an existing domain as new (or vice versa)?

A discussion on these failure cases, along with possible mitigation strategies, would strengthen the paper’s robustness.

**Relation To Broader Scientific Literature:**

N/A

**Theoretical Claims:**

NO theoretical claim

---

> ### Author Rebuttal · Authors · 2025-03-31
>
> We sincerely thank the reviewer for their positive feedback and insightful questions. We appreciate the thorough review and the recommendation to accept our paper. Below, we address the specific questions and concerns raised:
> ## Q1. Memory Efficiency Analysis
> In Lines 216–219, we compare with [1] where test samples are directly stored in a buffer. Our approach stores only test batch statistics, requiring significantly less space (e.g., from 64×3×224×224 per batch to just 2×768, about 0.02% of the original size). Additionally, our memory analysis on ImageNet-C using a single RTX 6000 Ada GPU:
> |Algo.|Memory (MB)|Err Rate(%)|
> |-|-|-|
> |Tent|3,877|51.0|
> |CoTTA|8,256|54.8|
> |ViDA|5,728|43.4|
> |Ours|3,879|39.9|
>
> DPCore achieves minimal memory footprint by 1) storing only learned prompts (0.34 MB total) and compact statistics, 2) requiring no test sample storage, and 3) sharing core elements across similar domains.
> ## Q2. Sensitivity Analysis of ρ Across Datasets
> We've analyzed ρ sensitivity across datasets, with results in this [figure](https://anonymous.4open.science/r/DPCore-Supp-8D17/ablation_rho_all.png). DPCore maintains stable performance on ImageNet-C, CIFAR10-C, and CIFAR100-C for ρ values between 0.7 and 0.9. We fix ρ=0.8 for all main experiments.
> ## Q3. Potential Failure Cases
> This is an excellent question that helps us better understand the limitations of our approach. However, "Overlapping Distributions", "Noisy or Small Domain Shifts" and "Inconsistent Weighting Effects" are not necessarily failure cases for our method. The goal of DPCore is not to learn an identical number of prompts as the number of domains or to identify all different domains. Instead, DPCore aims to update the same prompt for similar domain groups. Specifically:
> 1. This design is motivated by our findings in Sec 3.3 and Appendix D.1, where we observed that a prompt learned on one domain (e.g., Gaussian Noise) could work effectively on a similar domain (e.g., Shot Noise), and performance could be further improved by slightly updating the prompt on the second domain.
> 2. Since distance-induced weights are used to generate the weighted prompt, a new domain can be effectively represented as a decomposition of existing domains. The weighted prompt might perform well on a new domain even when derived from different domains, which is acceptable in our framework since domain similarity is evaluated by distance and prompts are learned through distance minimization.
>
> One potential failure case for our method would be when each test batch contains data from multiple domains. In this scenario, the statistics of each batch would not be stable, and our method might treat each batch as a different domain. This would reduce efficiency since the algorithm would need to learn prompts for each test batch from scratch, similar to what we show in Table 3 Exp-2 (learning prompts from scratch for each batch but the entire batch comes from the same domain). This scenario would pose challenges to most CTTA methods, as most assume each test batch contains data from a single domain. We plan to explore this direction in future work.
> ## Q4. Memory Optimization for Prompt Coreset
> Regarding memory overhead concerns, maintaining prompts in memory is quite negligible in practice. For example, the 14 prompts learned for ImageNet-C domains only require 0.08M parameters (0.34MB). If we were to allocate the same number of parameters as ViDA (7.13M), we could store approximately 1,247 prompts, making DPCore highly memory-efficient in practice.
>
> While our current implementation has modest memory requirements, exploring potential optimizations aligns well with practical deployment considerations. We have explored two approaches to maintain a fixed-size coreset on ImageNet-C with K=15 (matching the number of domains). Once the number of prompts exceeds K, we either 1) discard the oldest prompt or 2) merge the most similar prompts (computing the average of the two prompts whose statistics distance is smallest). We evaluated these strategies in the 10-different-order setting (Fig.3c) and report the average error rates:
> |Algo.|Err Rate(%)|
> |-|-|
> |Source|55.8|
> |DPCore (flexible K)|40.2|
> |DPCore (K=15, discard)|42.3|
> |DPCore (K=15, merge)|41.1|
>
> Both strategies still significantly improve upon the source model, though merging existing prompts performs slightly better than simply discarding one of them.
>
> In real-world scenarios, the number of domains is typically unknown, and fixing K at a small value could lead to suboptimal performance. Since the coreset grows only when encountering unseen domains and the memory overhead is negligible, our default approach is to allow K to evolve naturally as adaptation progresses.
>
> We appreciate the reviewer's thoughtful questions, which have helped us articulate our approach's strengths and limitations. We believe addressing these points will further strengthen the paper.
>
> [1] "Learning to Prompt for Continual Learning" CVPR 2022

---

### Official Review · Reviewer_koCY · 2025-03-14

**Overall Recommendation:** 3

**Summary:**

This paper utilizes a dynamic prompt coreset (DPCore) for continual test-time adaptation (CTTA). DPCore involves three components: visual prompt adaptation, prompt coreset, and a dynamic update mechanism for either updating the existing or, creating new prompts based on how similar the prompt is to the ones in the corset.
Experiments on benchmark datasets have been reported, showing better performance compared to existing approaches.

**Claims And Evidence:**

Mostly yes. Refer to Weaknesses and Questions.

**Essential References Not Discussed:**

NA

**Experimental Designs Or Analyses:**

Refer to Weaknesses and Questions.

**Methods And Evaluation Criteria:**

Mostly yes. Refer to Weaknesses and Questions.

**Other Comments Or Suggestions:**

In Table 3, gain for Exp-3 should be +10.7 instead of +8.7.

**Other Strengths And Weaknesses:**

**Strengths**
* An interesting idea of updating an existing prompt or adding a new prompt to the coreset based on the ratio of distances with or without the weighted prompt.
* Experimental results show improvement over existing approaches over compared benchmarks.

**Weaknesses**
* The requirement of unlabeled source example data (even if only 300), violates the source data-free assumption.
* CCN architecture experimental results on widely used benchmark is lacking (refer to Questions)

**Questions For Authors:**

1. How does the proposed approach perform on CIFAR10C, CIFAR100C, and ImageNetC on CNN-based architecture as discussed in the continual test-time adaptation line of work such as [1], [2], [3]?
2. Is the idea of dynamically updating coresets for prompt novel? Or is the paper utilizing this idea from an existing work that explored it for a different problem?

**References**
1. Wang, Qin, et al. "Continual test-time domain adaptation." Proceedings of the IEEE/CVF Conference on Computer Vision and Pattern Recognition. 2022.
2. Song, Junha, et al. "Ecotta: Memory-efficient continual test-time adaptation via self-distilled regularization." Proceedings of the IEEE/CVF Conference on Computer Vision and Pattern Recognition. 2023.
3. Brahma, Dhanajit, and Piyush Rai. "A probabilistic framework for lifelong test-time adaptation." Proceedings of the IEEE/CVF Conference on Computer Vision and Pattern Recognition. 2023.

**Relation To Broader Scientific Literature:**

The key contribution of the paper is that it enhances the TTA ability of a model. However, the contributions are limited to continual TTA and do not generalize much beyond it.

**Theoretical Claims:**

I have read the theorems in the Appendix, but not checked their proof in detail.

---

> ### Author Rebuttal · Authors · 2025-03-31
>
> We sincerely thank the reviewer for their careful reading of our paper and constructive feedback. Below, we address the specific concerns and questions raised:
> ## Q1. Source Data Requirement
> We appreciate this concern and would like to clarify:
>
> 1. DPCore requires source data only before adaptation starts. In practice, feature statistics could be provided by the model publisher alongside the pre-trained model. Moreover, several SOTA methods (e.g., ViDA, EcoTTA, VDP, DePT, BeCoTTA) also require source data for preparation. The key difference is that these methods need to warm-up their parameters on the **entire** source dataset for several epochs, which demands significantly more data and computation than our approach. They require both forward and backward propagation on the entire labeled dataset, whereas our method needs only a single round of forward passes on 300 unlabeled examples to extract features.
> 2. When no statistics are provided during preparation, they could be computed from the test data stream. In real-world scenarios, ID data is usually mixed with OOD data in the test batch stream. Filtering out 300 ID samples is straightforward using simple metric thresholds such as entropy.
> 3. As demonstrated in Fig.5c, our method's performance remains stable even with as few as 50 unlabeled source examples.
> 4. Most importantly, we show in Fig.5c and Appendix F.2 that DPCore can function effectively without **ANY** source data by using public datasets (e.g., STL10) as proxy reference data. Even in this extreme scenario, DPCore still outperforms the source model by +10.2% on ImageNet-C.
>
> Therefore, while we technically use source examples during preparation, our approach is source-data free during test time and can even operate without source data entirely, making it practical for real-world scenarios with limited or no access to source data.
> ## Q2. Performance on CNNs
> While our paper primarily focused on ViTs due to their strong representation capabilities, we have conducted additional experiments on ResNet for the datasets mentioned. The choice of models for each dataset follows EcoTTA. For CNNs, instead of learning visual prompts, the coreset stores NormLayer parameters (same approach as Table 3 Exp-3):
> |Dataset|Model|Source|Tent|CoTTA|EcoTTA|PETAL|Ours|
> |-|-|-|-|-|-|-|-|
> |ImageNet-C|ResNet50|82.4|66.5|63.2|63.4|62.7|**61.0**|
> |CIFAR10-C|WideResNet-28|43.5|20.7|16.3|16.8|16.0|**15.7**|
> |CIFAR100-C|WideResNet-40|69.7|38.3|38.1|36.4|36.8|**35.4**|
>
> These results demonstrate that DPCore consistently outperforms existing methods across all CNN architectures and datasets, confirming our approach's effectiveness beyond ViT architectures.
> ## Q3. Novelty of Dynamic Prompt Coreset
> The dynamic prompt coreset approach in our paper is indeed novel for test-time adaptation. While coresets have been explored in various machine learning tasks (e.g., continual learning), their application to managing prompts in TTA scenarios has not been previously investigated. The key novelty lies in:
> 1. The design of a coreset specifically for storing and managing visual prompts in dynamic CTTA. Prior methods continuously update the same parameters across different domains, which leads to convergence issues with brief domain exposures, risks forgetting previously learned knowledge, or misapplies it to irrelevant domains. DPCore manages domain knowledge through a dynamic prompt coreset to mitigate these issues.
> 2. The dynamic coreset approach is not restricted to prompts but can also be used for other types of parameters such as NormLayer parameters. It still improves the performance of the source model by +10.7% when applied to NormLayer parameters (Table 3 Exp-3). These results show that our dynamic coreset can be applied to other model architectures beyond ViT and integrated with other TTA methods.
> 3. Our novel decision mechanism determines whether to update existing prompts or create new ones based on domain similarity, mitigating negative transfer between dissimilar domains while allowing coreset prompts to be updated on similar ones. This approach leverages domain similarities through weighted prompt generation, computing efficient combinations that ensure constant evaluation time regardless of coreset size.
>
> Related work such as [1] explores prompt management for supervised continual learning, but requires labeled data and addresses a fundamentally different problem (discussed in lines 216–219). Similarly, online K-means clustering has been used in various applications, but our adaptation of these ideas to prompt-based CTTA with a dynamic update mechanism is novel.
> ## Others
> We will correct the Table 3 error (gain for Exp-3 should be +10.7, not +8.7) in the camera-ready version. The code will be published upon acceptance.
>
> We appreciate the thorough review and constructive feedback and hope these clarifications address all concerns.
>
> [1] "Learning to Prompt for Continual Learning" CVPR2022

---

> > ### Comment · Reviewer_koCY · 2025-04-07
> >
> > Thank you to the authors for responding.
> > I don't have any other questions or remarks.
> >
> > Thanks.

---

> > > ### Author Response · Authors · 2025-04-07
> > >
> > > Dear Reviewer koCY,
> > >
> > > We are happy to hear that our rebuttal addressed your concerns. We promise to open-source the code and incorporate the corrected Table 3 and the ResNet results in the camera-ready version.
> > >
> > > If you need any further clarifications or experiments, we're happy to provide them. Given that your questions have been addressed satisfactorily, we would be grateful if you might reconsider your score/rating.
> > >
> > > Thank you for your time and thoughtful review.
> > >
> > > Best regards,
> > >
> > > Authors

---

### Official Review · Reviewer_92up · 2025-03-15

**Overall Recommendation:** 3

**Summary:**

The paper focuses on continual test-time adaptation. Under a more complex setting where domains recur with varying frequencies and durations, the paper proposes DPCore with a dynamically updated prompt coreset for the adaptation on different distributions. The experiments on both common continual test-time adaptation and the proposed dynamically continual test-time adaptation settings demonstrate the effectiveness of the proposed method.

**Claims And Evidence:**

Most of the claims made in the submission are clear with evidence.

However, as there are already many different settings for continual test-time adaptation [1, 2], it is not clear why the newly proposed setting (CDC) is unique, more challenging, and more like real-world applications. More discussions and evidence are required for the setting.

**Essential References Not Discussed:**

N/A

**Experimental Designs Or Analyses:**

The experiments include both the common continual test-time adaptation setting and the newly proposed setting. Results and ablation studies demonstrate the effectiveness of the proposed method.

However, there are still some questions about the experiments.

1. One question is whether the hyperparameters of the baseline methods in Table 4 are the same as the proposed method (e.g., batchsize = 64, 300 available source samples at test time). Because SAR (Niu et al., 2023) can achieve 43.7% error rate with batchsize=1 in their table 4, which is even lower than the number reported in this paper (45.6%). I assume that with a larger batch size, the results of SAR will be better.

2. The previous methods also propose some different challenging settings for continual test-time adaptation, for example, the "imbalanced label shifts", "mixture of different corruption types", and "with batch size=1" in SAR (Niu et al., 2023). It looks like the proposed method doesn't perform well with "batchsize=1". Is there any analysis for the failure case? Can the proposed method handle the other two challenging settings?

**Methods And Evaluation Criteria:**

Overall, the proposed method and evaluation criteria sound good.

However, some details of the method are not clear and are confusing.

1. Visual prompt tuning has already been investigated by several papers and there are different strategies to introduce visual prompt into the transformer. How does the method decide the position and layers of the learnable prompt? Is there any experiment to demonstrate the strategy? How to decide the positional embedding of the learnable prompt?

2. Typically test-time adaptation use entropy minimization for optimization, while the proposed method aligning the adapted target features with source features? What's the motivation and benefits to use such objective function?

3. The proposed method use the feature statistics to generate prompts for new domains. However, the output features (statistics) and the input prompts are in different feature spaces. How can the method guarantee that the relationship (or similarities) between statistics can reflect the relationships between prompts?

**Other Comments Or Suggestions:**

Commonly, the related works should be included in the main paper for the readers to understand the task and previous methods.

**Other Strengths And Weaknesses:**

Strengths:

1. The paper is well-written and easy to follow.

2. The experiments demonstrate the effectiveness of the proposed method

Weaknesses:

1. It is not clear why the newly proposed settings for continual test-time adaptation is unique, challenging, and practical compared with the previous methods or settings.

2. Some details of the method are not clearly stated and analyzed. Please refer to "Methods And Evaluation Criteria"

3.  The comparisons and evaluations also need more details and analyses. Please refer to "Experimental Designs Or Analyses"

4. Since the similarities are calculated between statistics without learnable prompts, each test batch requires two feedforward passes of the model for prompt generation and one more backpropagation for optimization. This will introduce much more computational costs, which is not so efficient.

**Questions For Authors:**

N/A

**Relation To Broader Scientific Literature:**

The paper is related to test-time adaptation and generalization problems, as well as continual learning.

**Theoretical Claims:**

The paper provides some theoretical analyses on how DPCore correctly assigns batches to respective clusters.

---

> ### Author Rebuttal · Authors · 2025-03-31
>
> We thank the reviewer and address each point:
> ## Q1. Value of Proposed CDC Setting
> Our CDC setting models real-world scenarios with irregular distribution shifts where domains recur unpredictably (Fig.1), unlike existing CTTA approaches that assume uniform changes. For example, an autonomous vehicle driving through a mountainous area would encounter brief tunnels, extended sunny stretches, and intermittent fog or rain with unpredictable patterns and durations. SOTA methods degrade significantly in CDC (ViDA's error rate increases from 43.4% to 52.1%), highlighting that CDC presents new challenges not adequately addressed by existing settings. CDC amplifies three critical limitations in existing settings: convergence issues with brief domains, catastrophic forgetting with irregular patterns, and negative transfer between dissimilar domains.
>
> These findings suggest CDC better reflects real-world challenges and provides a rigorous testbed for evaluating CTTA methods. Multiple reviewers recognized our approach: te3N noted CDC is "**practical and aligns well with real-world dynamic changes**" and "**the proposed new setting is valuable**" while jKst called it "**novel and interesting**." These independent assessments validate our motivation. Differences between CDC and existing settings (e.g., SAR) are discussed in Appendix E.2.
> ## Q2. Visual Prompt (VP) Implementation Details
> We follow standard shallow VP implementation from [1], prepending prompt tokens to image tokens after the CLS token (Fig.2). The CLS token is invariant to the location of prompts since they are inserted after positional encoding ([1]). We explored using entropy minimization to learn VP at test time before settling on distribution alignment, but it led to degenerate solutions (increasing error rate by ~10% over the source model), which aligns with findings in recent work ([2]).
>
> It's important to note that our main contribution is not how to learn VP, but rather DPCore's ability to manage domain knowledge through a dynamic coreset for CTTA. Our method remains effective when VP is replaced with NormLayer parameters (Table 3 Exp-3). For additional details, please see our response to Reviewer jKst's Q2.
> ## Q3. Relationship between Feature Statistics and Prompts
> While feature statistics and prompts exist in different spaces, our objective function (Eq.5) creates a meaningful mapping between them through optimization. When we learn a prompt to minimize the distance between source and target statistics, we're effectively creating a prompt space that aligns with the statistical relationships in feature space. Our empirical evidence supports this approach: Table 5 shows prompts learned for similar domains transfer effectively between them; Fig.3a demonstrates our prompts consistently reduce domain gaps across all corruption types; and Fig.3c confirms stability across diverse domain orders. Essentially, the optimization creates a reliable bridge between statistical relationships and prompt functionality.
> ## Q4. Comparison with SAR
> There's a misunderstanding in the comparison. SAR addresses **single** domain (evaluating each domain independently) while we focus on **continually changing** domains (15 domains as one sequential task). SAR isn't designed for CTTA, so its performance drops. We use consistent hyperparameters (e.g., batch size=64) across all methods for fair comparison (detailed in Appendix B).
>
> The three challenging settings in SAR are originally designed for **single** domain, not **changing** CTTA scenarios. But we further address these for CTTA in Appendix E.3 and F.1:
> 1. **Batch size=1**: Fig.5d shows all CTTA methods struggle with small batches in changing domains. Our DPCore-B variant (Appendix F.1) addresses this by using a negligible buffer to accumulate sample features, achieving 41.2% error rate with single-sample batches (vs. 39.9% with batch size=64).
> 2. **Imbalanced label shifts**: Though our method wasn't specifically designed for this challenge, it achieves better performance in this setting (improving the error rate to 43.9% as shown in Table 10).
> 3. **Mixed domain**: In the CTTA setting, batches containing data from different domains usually occur near domain boundaries. We verified our method in this case with DPCore-B, which can still achieve an improvement of +14.6% over the source model (Table 11).
> ## Q5. Computational Efficiency Concerns
> We have carefully analyzed our method's computational efficiency (Table 4 and Appendix F.3). While DPCore does require additional computation compared to some simpler methods (e.g. Tent), it remains significantly more efficient than many SOTA approaches (e.g., CoTTA, ViDA). Our method requires a similar number of backpropagation operations (~1) but significantly fewer forward propagations (3.1) than CoTTA (11.7) and ViDA (11.0), which require forwarding extra augmented test data.
>
> [1] "Visual Prompt Tuning" ECCV2022
>
> [2] "Test-Time Model Adaptation with Only Forward Passes" ICML2024

---

> > ### Comment · Reviewer_92up · 2025-04-09
> >
> > I thank the authors for the rebuttal, which solves most of my concerns. Now I tend to weak accept of the paper.

---

### Decision · Program_Chairs · 2025-05-01

**Decision:**

Accept (poster)

**Comment:**

This paper introduces a new TTA framework called DPCore that managed domain knowledge through a coreset of prompts updated dynamically during test, as well as a more challenging test-time adaptation (TTA) scenario, dubbed continual dynamic change (CDC). The proposed method demonstrated outstanding performance on both conventional and CDC settings.

The reviewers recognized the intriguing idea of managing and exploiting the dynamic prompt coreset, extensive experiments, strong performance, and clarity of the paper. However, they at the same time raised concerns with unclear motivation and contribution of the CDC setting (92up), lack of details, motivation, and justification for the proposed method (92up), insufficient details and analysis in evaluations of baseline models (92up), violating the fully test-time adaptation assumption (koCY), missing analysis on memory-efficiency (te3N), potential sensitivity to extra hyperparameters (te3N, jKst), and questionable effectiveness of the updating mechanism (jKst). The rebuttal and subsequent discussions addressed most of these concerns and in the post-rebuttal phase, all reviewers except one turned towards acceptance. The negative reviewer did not come back, but the AC found that his/her remaining concern on the updating scheme has been addressed successfully by the last comment by the authors.

The AC found that the strengths of the paper and the rebuttal outweigh the remaining concerns, and thus recommend acceptance of the paper. The authors are strongly encouraged to carefully revise the paper to reflect the valuable comments by the reviewers and to add new results brought up in the rebuttal and discussions.